# Splicing repression allows the gradual emergence of new Alu-exons in primate evolution

Jan Attig[1,2†], Igor Ruiz de los Mozos[1†], Nejc Haberman[1†], Zhen Wang[3], Warren Emmett[1,4†], Kathi Zarnack[5], Julian König[6*], Jernej Ule[1,2*†]

[1]Department of Molecular Neuroscience, UCL Institute of Neurology, London, United Kingdom; [2]MRC-Laboratory of Molecular Biology, Cambridge, United Kingdom; [3]Institute de Biologie de l'ENS (IBENS), CNRS UMR 8197, Paris, France; [4]University College London Genetics Institute, London, United Kingdom; [5]Buchmann Institute for Molecular Life Sciences (BMLS), Goethe University Frankfurt, Frankfurt, Germany; [6]Institute of Molecular Biology (IMB), Mainz, Germany

*For correspondence: J.Koenig@ imb-mainz.de (JK); j.ule@ucl.ac.uk (JU)

Present address: †The Francis Crick Institute, London, United Kingdom

**Abstract** Alu elements are retrotransposons that frequently form new exons during primate evolution. Here, we assess the interplay of splicing repression by hnRNPC and nonsense-mediated mRNA decay (NMD) in the quality control and evolution of new Alu-exons. We identify 3100 new Alu-exons and show that NMD more efficiently recognises transcripts with Alu-exons compared to other exons with premature termination codons. However, some Alu-exons escape NMD, especially when an adjacent intron is retained, highlighting the importance of concerted repression by splicing and NMD. We show that evolutionary progression of 3' splice sites is coupled with longer repressive uridine tracts. Once the 3' splice site at ancient Alu-exons reaches a stable phase, splicing repression by hnRNPC decreases, but the exons generally remain sensitive to NMD. We conclude that repressive motifs are strongest next to cryptic exons and that gradual weakening of these motifs contributes to the evolutionary emergence of new alternative exons.

## Introduction

Retrotransposed elements account for ~45% of the human genome (*Deininger and Batzer, 2002*). The primate-specific *Alu* elements are the second most prevalent class with almost 1.2 million copies, and thousands of them can be spliced to create Alu-exons (*Lev-Maor et al., 2003*; *Sorek et al., 2004*; *Zarnack et al., 2013*). Only few Alu-exons encode for novel protein isoforms (*Lin et al., 2016*), and for several, the evolutionary history of their exonisation has been described (*Krull et al., 2005*; *Moller-Krull et al., 2008*; *Singer et al., 2004*). Alu elements are particularly prone to exonisation because as few as three single nucleotide mutations from the Alu consensus sequence are sufficient to create cryptic 5' and 3' splice sites (*Lev-Maor et al., 2003*; *Sorek et al., 2004*), and the left-arm Alu sequence contains a CUAUU sequence that can serve as branchpoint (*Mercer et al., 2015*). Annotated Alu-exons are usually alternatively spliced and show low inclusion levels (*Sorek et al., 2002*). Although it is clear that the presence of splice sites and other positive splicing elements was crucial for the emergence of Alu-exons (*Lev-Maor et al., 2003*; *Sorek et al., 2004*), the role of repressive splice elements is less well understood. Like other types of cryptic exons, Alu-exons often contain detrimental sequences that can lead to the production of misfolded or dominant negative protein variants and are therefore associated with many human diseases (*Kaneko et al., 2011*).

Thus, it is important to understand the protective molecular mechanisms imposing constraints on the emergence and expression of Alu-exons.

Virtually, all human genes contain an Alu element in at least one intron. Alu elements require a polyA-tail for their retrotransposition (*Doucet et al., 2015*), and therefore, when they insert into other genes in an antisense orientation, they start with a polyuridine tract (U-tract). Moreover, these antisense Alu elements often contain cryptic splice sites (*Lev-Maor et al., 2003*). While U-tracts can allow 3' splice site recognition in vitro (*Bouck et al., 1998*), we previously showed that the RNA-binding protein heterogeneous nuclear ribonucleoprotein C1/C2 (hnRNPC) binds these U-tracts in vivo to block recruitment of the spliceosomal factor U2 auxiliary factor 65 kDa subunit (U2AF2, also U2AF65). In principle, it is plausible that other proteins with known preference for U-rich motifs (e.g. HuR, TIA, TDP43) also bind the Alu U-tract and repress Alu exonisation; and indeed HuR and TDP43 show enriched binding at antisense Alu elements (*Kelley et al., 2014*). Yet, depletion experiments for TDP43, TIA, HuR, PTB and hnRNPA1 did not show increased Alu-exon inclusion in their absence (*Zarnack et al., 2013*; *Kelley et al., 2014*). Thus, the U-tract:hnRNPC interaction is crucial to prevent the splicing machinery from accessing cryptic splice sites at Alu-exons.

Our previous study of hnRNPC depletion uncovered exonisation of more than 1900 Alu elements (*Zarnack et al., 2013*). However, the total number of Alu-exons regulated by hnRNPC is likely to be even larger, since Alu-exon-containing transcripts (Alu-exon transcripts) may evade detection if they are unstable. For instance, the presence of inverted Alu repeats within 3' untranslated regions (3' UTRs) causes nuclear retention (*Chen and Carmichael, 2009*; *Chen et al., 2008*). Moreover, most Alu-exons will introduce a premature termination codon (PTC) into the transcript, and Alu-exon transcripts are therefore likely to be targeted by nonsense-mediated mRNA decay (NMD).

Here, we set out to investigate the importance of hnRNPC-mediated splicing repression and NMD-dependent mRNA surveillance in the quality control of Alu-exon transcripts. This identified 3101 new Alu-exons that are normally repressed by one or both pathways in wild-type cells. Upon hnRNPC but not UPF1 depletion, many Alu-exons are included together with flanking intronic sequences. Surprisingly, these intron-retaining Alu transcripts are generally resistant to NMD, even though they harbour PTCs, are exported to the cytoplasm and found in association with polysomes. Analysing the evolutionary paths towards exonisation, we find that evolutionary emergence of a stronger 3' splice site in an exonising Alu element is coupled with a longer repressive U-tract. At the same time, splicing repression by hnRNPC is decreased at ancient Alu elements, while they remain sensitive to NMD. We conclude that Alu-exon formation proceeds through distinct evolutionary stages that rely on complementary repressive mechanisms.

## Results

### Known Alu-exons correlate with decreased gene expression

Many intronic Alu elements in the human genome have acquired mutations leading to the formation of cryptic splice sites (*Lev-Maor et al., 2003*; *Sorek et al., 2004*). In the UCSC gene annotation, 2,657 Alu-exons are present (human genome version GRCh37/hg19), for which at least one of the splice sites originates from an Alu element. However, this is a relatively small number, given that Alu elements constitute 12.2% of the intronic sequence of protein-coding genes. In contrast, only 4% of exonic sequence derives from Alu elements. Notably, the contribution of Alu element sequences is less than 0.6% in constitutive exons, indicating a strong bias towards alternative inclusion of Alu-exons. This suggests that the formation of alternative and constitutive Alu-exons is under tight control.

To characterise whether Alu-exons have an impact on gene expression, we compiled a set of 4293 known Alu-exons, including 574 and 2083 annotated as constitutive and alternative exons, respectively, as well as 1636 cryptic Alu-exons that we found in hnRNPC-depleted HeLa cells (*Zarnack et al., 2013*). We then analysed the expression of protein-coding genes (referred to as Alu-exon or non-Alu genes) across 16 human tissues using the Illumina BodyMap 2.0 RNAseq data. In all tissues, Alu-exon genes with an alternative or a constitutive Alu-exon showed a clear trend to be less expressed than genes without Alu-exons, and this was most pronounced for genes containing a constitutive Alu-exon (*Figure 1A* and *Figure 1—figure supplement 1*).

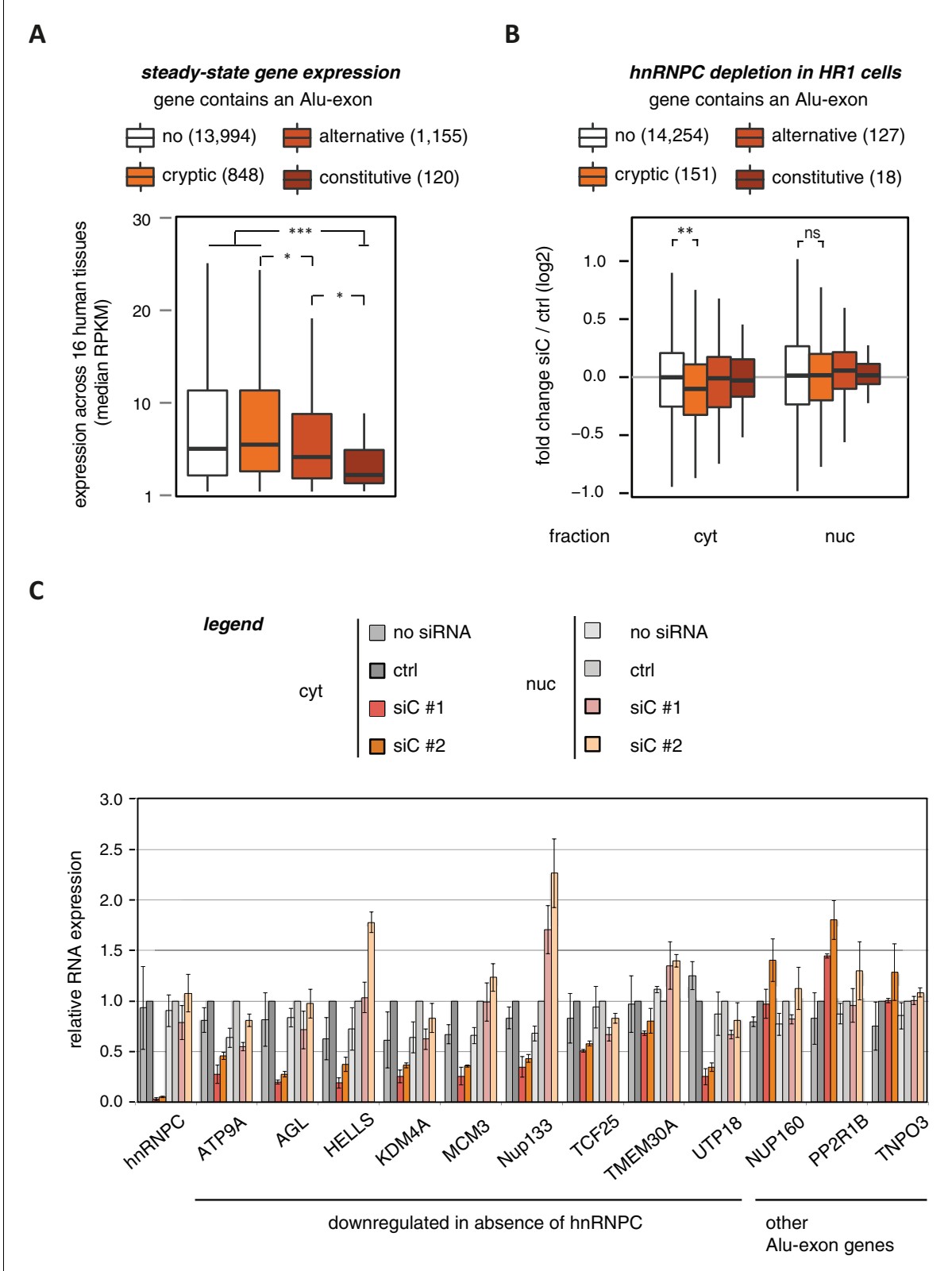

**Figure 1.** Alu-exons are associated with reduced gene expression. (**A**) Expression levels of genes containing a cryptic, alternative or constitutive Alu-exon across a panel of 16 human tissues. For each gene, the median expression level was calculated from the Illumina BodyMap 2.0 dataset, considering only widely expressed genes (median RPKM ≥ 1). Expression in each individual tissue is shown in *Figure 1—figure supplement 1*. Differences in the distribution were tested by an ANOVA design, for details see Materials and methods. (**B**) RNAseq data of cytoplasmic and nuclear

*Figure 1 continued*

RNA from HR1 cells transfected with siRNAs against hnRNPC (siC #1 and #2) or controls ('no siRNA': mock transfection; ctrl: unspecific control oligonucleotide) was used to test if inclusion of Alu-exons impacts on gene expression. Protein-coding genes with or without cryptic, alternative or constitutive Alu-exons were grouped, considering only Alu-exons with sufficient coverage (minimal five reads/million including one junction-spanning read; 550 Alu-exons). Shown is the distribution of expression fold changes (log$_2$). To estimate the expression fold changes, we compared cells depleted of hnRNPC (siC #1 and #2) against controls (no siRNA and control oligonucleotide) using DESeq (*Anders and Huber, 2010*). Differences between Alu-exon gene abundance across groups were tested by Kruskal-Wallis Rank Sum test within each RNA fraction (cytoplasmic RNA, p-value = 0.01819; nuclear RNA, p-value = 0.3646). Indicated pairwise comparisons were tested with a two-sided Wilcoxon Rank Sum test. ** indicates p-value < 0.01. (**C**) RNA expression levels of 12 Alu-exon genes identified from the cytoplasmic and/or nuclear RNAseq datasets. Loss of transcripts from Alu-exon genes in hnRNPC-depleted cells was generally restricted to cytoplasmic RNA. Gene expression was quantified by quantitative RT-PCR from HR1 cells transfected with siRNAs against hnRNPC (siC #1 and #2) or controls ('no siRNA': mock transfection; ctrl: unspecific control oligonucleotide), error bars represent standard deviation of the mean (s.d.m.; n = 3 or 4). *Figure 1—figure supplement 2* shows the expression of Alu-exon genes in the individual tissues, which is summarised in *Figure 1A*. *Figure 1—figure supplement 2* provides quality control on the nuclear-cytoplasmic fractions by Western blot and quantitative RT-PCR. *Figure 1—figure supplement 3* presents semi-quantitative RT-PCR quantifications of individual Alu-exon transcripts compared to the Alu-exon-free transcripts. This validates that Alu-exon transcripts are depleted from the cytoplasmic RNA pool. *Figure 1—figure supplement 4* demonstrates that cytoplasmic depletion of Alu-exon transcripts is not the result of a lack of mRNA export from the nucleus.

The following figure supplements are available for figure 1:

**Figure supplement 1.** Alu-exon gene expression in various human tissues.
**Figure supplement 2.** Quality control of cytoplasmic and nuclear RNA fractions.
**Figure supplement 3.** Alu-exon transcripts are frequently depleted from the cytoplasmic RNA pool.
**Figure supplement 4.** Cytoplasmic depletion of Alu-exon genes is not caused by a defect in mRNA export upon hnRNPC depletion.

The reduced expression of Alu-exon genes might result from a combination of two scenarios. Genes with low expression might be less essential to the organism, or evolve more quickly, which would decrease negative selection against inclusion of Alu-exons. Alternatively, inclusion of Alu-exons could be the primary cause of the low gene expression as a result of quality control pathways that degrade Alu-exon transcripts.

## Inclusion of cryptic Alu-exons reduces cytoplasmic expression of the Alu-exon containing transcripts

Since annotated Alu-exons are associated with reduced gene expression across tissues, we wondered whether inclusion of cryptic Alu-exons could similarly diminish expression of their host genes. Cryptic Alu-exons are not included in the presence of hnRNPC, since splicing of Alu-exons is strongly repressed by hnRNPC (*Zarnack et al., 2013*). Hence, we set up hnRNPC depletion experiments, in which we obtained robust depletion of hnRNPC within 48 hr after transfection of two independent siRNAs (siC #1 and siC #2, *Figure 1—figure supplement 2A*), and biochemically separated cytoplasmic and nuclear RNA (*Bhatt et al., 2012*; *Bühler et al., 2002*). Using Western blots and quantitative RT-PCR (qPCR), we observed no cross-contamination between fractions at the protein level and an almost complete absence of introns in the cytoplasmic RNA fraction (*Figure 1—figure supplement 2B C*). We performed these experiments in HR1 cells, a HEK293 derivate with a 4-hydroxytamoxifen (4-HT)-inducible RAF1 kinase transgene allowing robust activation of the ERK MAPK pathway in less than 30 min (see below; *Figure 1—figure supplement 4B*, *Ewings et al., 2007*). The resulting ERK-inducible gene expression enabled us to specifically follow newly transcribed RNAs in our analysis. RNA was collected at 0, 30 and 60 min after RAF1 kinase induction from cells transfected with the different siRNAs. RNA was depleted of ribosomal RNA and converted into unstranded RNAseq libraries, for which we sequenced a total of ~103 million uniquely mapping reads.

Using DESeq (*Anders and Huber, 2010*), we found 1094 genes to be differentially expressed in either the cytoplasmic or the nuclear RNA fractions upon hnRNPC depletion (adjusted p-value < 0.01). The differentially expressed genes are highly enriched for Alu-exons (*Figure 1—figure supplement 2D*, p-value < $2.2e^{-16}$, Fisher's exact test). As predicted, only genes with cryptic Alu-exons showed a

decrease in expression after hnRNPC depletion, while genes with annotated Alu-exons did not (*Figure 1B*). Moreover, cryptic Alu-exons were associated with decreased expression only in cytoplasmic but not nuclear RNA. We used qPCR to validate the cytoplasmic loss of expression for nine out of nine well-expressed Alu-exon genes (*Figure 1C*). We also validated the expression of three Alu-exon genes for which we did not detect hnRNPC-mediated changes in our RNAseq data (*Figure 1C*), presumably due to low inclusion levels (*NUP160, TNPO3*) or positioning of the Alu-exon in the 5' UTR (*PP2R1B*). Finally, we used semi-quantitative RT-PCR to monitor the relative abundance of the transcript isoforms containing the Alu-exon (Alu-exon transcripts). For two out of four genes (*AGL, TIMM23*), the Alu-exon transcripts were less abundant in the cytoplasm than in the nucleus (*Figure 1—figure supplement 3*), and *NUP133* showed more complex splicing patterns (see below). In summary, we conclude that the inclusion of Alu-exons followed by cytoplasmic loss of the Alu-exon transcripts frequently leads to reduced gene expression in response to hnRNPC depletion.

## The previously proposed mRNA export function of hnRNPC does not affect Alu-exon transcripts

A recent report described hnRNAs as necessary for the export of mRNAs (*McCloskey et al., 2012*), which could serve as a possible explanation for the cytoplasmic loss of Alu-exon transcripts upon hnRNPC depletion. However, we did not observe any change in the cytoplasmic nor nuclear mRNA content upon depletion of hnRNPC (*Figure 1—figure supplement 4A*). Within our time-course of 4-HT induction, hnRNPC depletion did not change the kinetics of cytoplasmic mRNA accumulation of the ERK-induced genes *FOS*, *ERG2* and *NDRG1* (*Saito et al., 2013*) (*Figure 1—figure supplement 4B*). Moreover, the vast majority of Alu-exon and other genes did not show an expression pattern suggestive of an mRNA export block (*Figure 1—figure supplement 4C*), nor did we detect an increased nuclear accumulation of mature RNAs in dependence of hnRNPC (*Figure 1—figure supplement 4D*). Hence, we observed neither a general nor an Alu-exon associated role of hnRNPC in mRNA export in our experiments. We therefore conclude that the cytoplasmic loss of Alu-exon transcripts under hnRNPC knockdown conditions in our study does not result from a nuclear export block, but instead might reflect their cytoplasmic degradation.

We noted that one of the Alu-exon genes, *NUP133*, encodes a protein of the nuclear pore complex that was shown to be required for mRNA export in *Xenopus laevis* oocytes (*Vasu et al., 2001*). We find that NUP133 protein levels continue to decrease at later time points beyond the 48 hr of hnRNPC depletion used in our experiments (*Figure 1—figure supplement 4E*), which might cause the previously described mRNA export defects in hnRNPC-depleted cells (*McCloskey et al., 2012*).

## NMD of Alu-exon transcripts reduces expression of the associated genes

Since Alu-exons commonly contain PTCs (*Sorek et al., 2002*), we next investigated whether their inclusion could target transcripts for NMD. For this purpose, we combined hnRNPC depletion with depletion of UPF1, a core factor of the NMD pathway (*Chan et al., 2007*; *Lykke-Andersen, 2000*). To systematically examine the impact of NMD on Alu-exon transcripts, we generated stranded RNA-seq libraries from HR1 cells depleted of hnRNPC, UPF1 or both (*Figure 2—figure supplement 1C*), which produced a total of 557 million uniquely mapping reads. As a control, we confirmed that UPF1 depletion led to up-regulation of known NMD targets (*Figure 2—figure supplement 1A and B*) and that among all exons detected in our RNAseq, PTC+ exons were significantly upregulated upon UPF1 depletion (*Figure 2—figure supplement 1D*)

Alu-exon inclusion increased upon depletion of hnRNPC, while the overall abundance of Alu-exon genes decreased. Notably, co-depletion with UPF1 restores expression of Alu-exon genes caused by loss of hnRNPC (*Figure 2A*). This strongly suggests that NMD is involved in the cytoplasmic loss of Alu-exon transcripts and that hnRNPC-mediated splicing repression and NMD act together to repress Alu-exon expression. Indeed, changes in exon abundance analysed by DEXSeq (*Anders et al., 2012*) showed that depletion of either hnRNPC or UPF1 alone resulted in increased Alu-exon abundance and that co-depletion of both factors showed a synergistic effect (*Figure 2B*). The Alu-exons most strongly repressed by hnRNPC and UPF1 caused the most pronounced decrease in gene expression upon hnRNPC depletion, verifying a quantitative relationship between exon inclusion and reduced transcript abundance (Spearman rank correlation, p-value < $2.2e^{-16}$). In

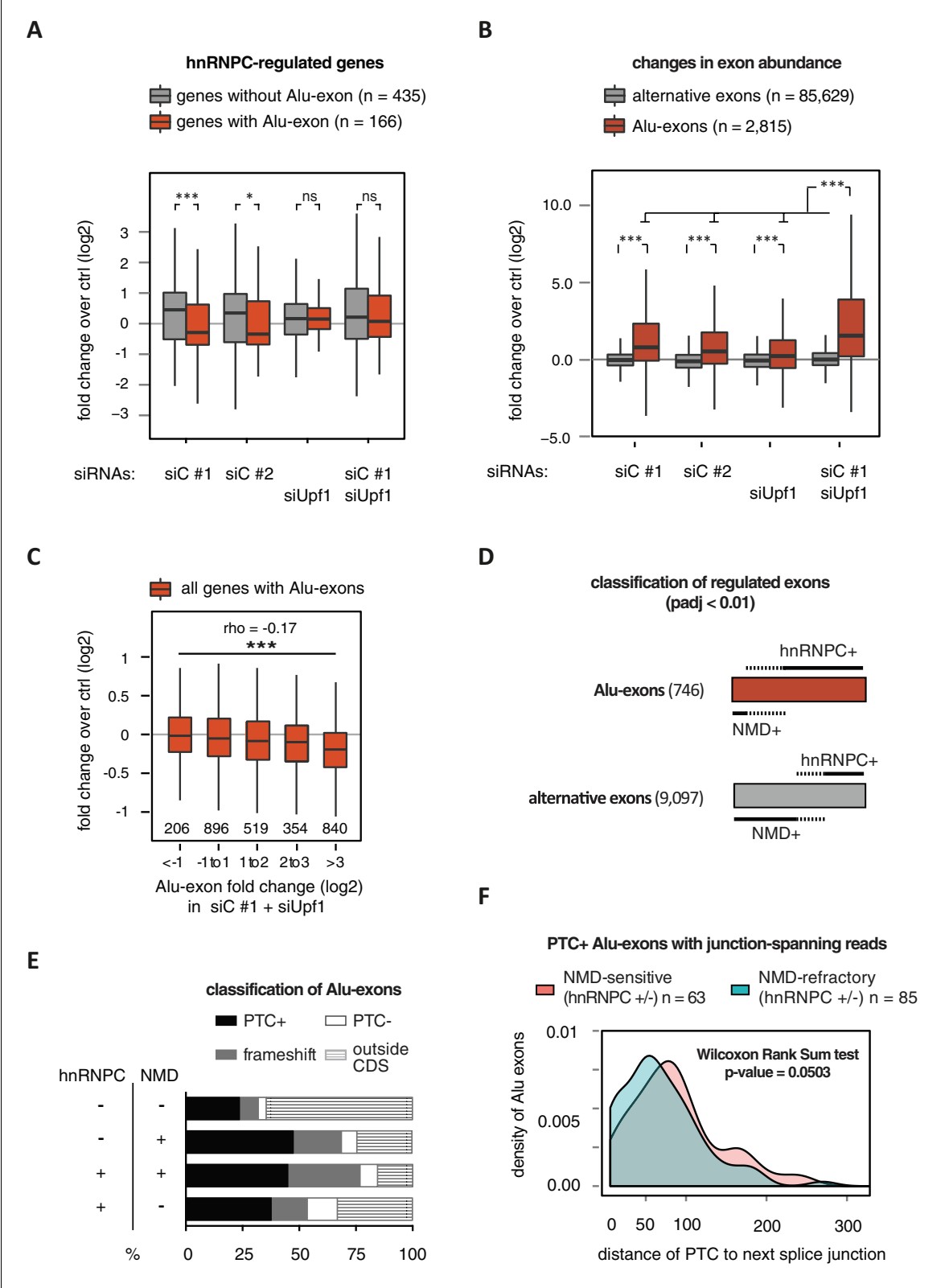

**Figure 2.** Alu-exons are a novel class of NMD substrates. (**A**) Depletion of hnRNPC in total RNA led to significant expression changes in 601 protein-coding genes (p-value < 0.01), of which 166 contain an Alu-exon. Boxplot showing the distribution of expression fold changes (log2) over control (unspecific control oligonucleotide) in cells depleted of hnRNPC (siC #1 and siC #2) and/or UPF1 (siUpf1) as indicated below. Differences between Alu-exon gene abundance across groups were tested by Kruskal-Wallis Rank Sum test (p-value < 2.2e⁻¹⁶), and pairwise comparisons within each condition

*Figure 2 continued on next page*

*Figure 2 continued*

were tested with a two-sided Wilcoxon Rank Sum test. * and *** indicate p-value < 0.05 and 0.001, respectively. (**B**) Boxplot showing the changes in exon abundance upon hnRNPC and/or UPF1 depletion as fold changes ($\log_2$) over control (unspecific control oligonucleotide), analysed by DEXSeq (*Lykke-Andersen et al., 2000*). Only alternative exons and Alu-exons in protein-coding genes were considered. Differences between Alu-exon abundance across groups were tested by Kruskal-Wallis Rank Sum test (p-value < $2.2e^{-16}$), and pairwise comparisons were corrected according to Siegel-Castellan. *** indicates p-value < 0.001. (**C**) Boxplot showing relationship between gene expression changes upon hnRNPC depletion and Alu-exon usage. Alu-exons were stratified according to their fold change ($\log_2$) in exon abundance in cells depleted of UPF1 and hnRNPC (siC #1), which should be the maximal achievable inclusion of each Alu-exon. Protein-coding genes with Alu-exons were filtered for a minimal RPKM of 1 (2809 genes in total). Significant anti-correlation between Alu-exon inclusion and Alu-exon gene expression levels was tested by Spearman correlation (indicated by $\boldsymbol{\rho}$ coefficient). (**D**) Depiction of the overlap of hnRNPC- and NMD-sensitive exons that are significantly regulated (adjusted p-value < 0.01) under the conditions in (**B**). Exon sets as in (**B**). Dashed lines visualise proportion of exons that are coordinately regulated by both hnRNPC and NMD. hnRNPC+ and NMD+ denote upregulation in hnRNPC-depleted and UPF1-depleted cells, respectively. All proportions are drawn to scale within each group. (**E**) Bar diagram showing the proportion of Alu-exons outside (grey hatching) and inside the coding sequence (CDS), with the latter being further subdivided into exons introducing a PTC (PTC+, black), without PTC but introducing a frame-shift (grey), or being in frame (PTC-, white) for the following Alu-exon categories: non-regulated (hnRNPC-/NMD-, n = 313), NMD-specific (hnRNPC-/NMD+, n = 72), shared-target (hnRNPC+/NMD+, n = 207), and hnRNPC-specific (hnRNPC+/NMD-, n = 465). (**F**) Density plot showing the distance of the PTC to the downstream splice site for NMD-sensitive (orange) and NMD-refractory (green) PTC+ Alu-exons. Only Alu-exons that were supported by junction-spanning reads on either side in our RNAseq data were taken into account. In *Figure 2—figure supplement 1*, the lack of functional NMD due to UPF1 depletion is validated by semi-quantitative and quantitative RT-PCR as well as by analysis of transcriptomic changes in RNAseq of UPF1-depleted cells. In *Figure 2—figure supplement 2*, we present semi-quantitative RT-PCR quantifications of individual Alu-exon transcripts compared to the Alu-exon free transcripts. This validates that Alu-exon transcripts are depleted from the cytoplasmic RNA pool by NMD.

The following figure supplements are available for figure 2:

**Figure supplement 1.** Validation of disenabling NMD by UPF1 depletion.

**Figure supplement 2.** Expression analysis of NMD-sensitive and NMD-refractory Alu-exons.

total, 746 of all Alu-exons are significantly repressed by either hnRNPC or UPF1 (adjusted p-value < 0.01).

The increased Alu-exon abundance upon co-depletion allowed us to re-evaluate the number of Alu elements which can exonise under different conditions. In HR1 cells, we identified a total of 5205 Alu-exons, including 3101 that were not identified in our previous study in HeLa cells (*Figure 2—figure supplement 1E*). Together with further Alu-exons that are annotated in UCSC but not expressed in either cell type, we found evidence for 6309 non-overlapping exonising Alu elements within 4243 human genes.

## PTC+ Alu-exons trigger NMD more efficiently than other PTC+ exons

Next, we compared the relative efficiency of hnRNPC and NMD in repressing Alu-exon transcripts. To this end, we assessed whether the increase in exon inclusion in co-depleted cells was mirrored in either of the single depletions. This enabled us to classify the 746 significantly repressed Alu-exons as 'sensitive' or 'refractory' to either factor, resulting in three groups of exons (*Figure 2D*): 72 NMD-specific Alu-exons (hnRNPC-/NMD+, 10% of the significantly regulated Alu-exons), 465 hnRNPC-specific Alu-exons (hnRNPC+/NMD-, 62%), and 207 shared-target Alu-exons (hnRNPC+/NMD+, 28%). For comparison, we used all Alu-exons that are expressed in HR1 cells but do not respond to either depletion (non-regulated, 313 Alu-exons) as well as all alternative exons of non-Alu origin that were significantly repressed (9097 alternative exons). We used semi-quantitative RT-PCR to validate the differential susceptibility to NMD for representative Alu-exons from each category (*Figure 2—figure supplement 2A and B*).

Intriguingly, approximately two-thirds of all hnRNPC-sensitive Alu-exons (hnRNPC-specific and shared-target) are refractory to NMD (*Figure 2D*), compared to one-third of alternative 'non-Alu' exons. To gain further insights, we computationally predicted which Alu-exons are located within the coding sequence (CDS) of the respective transcripts and introduce a PTC (PTC+, *Figure 2E*). Almost 85% of the shared-target Alu-exons are within the CDS, compared to only 35% of the non-regulated Alu-exons, suggesting that repression of Alu-exons is particularly important within the CDS. Of 175 shared-target Alu-exons within the CDS, 65% introduce a PTC and additional 21% a frame-shift,

underlining their potentially deleterious impact. However, we were surprised to find similar PTC+ proportions among the hnRNPC-specific target exons in the CDS that are refractory to NMD (56% with PTC plus additional 24% with a frame-shift). This suggested that many PTC-containing Alu-exons escape quality control by NMD. As a control, we repeated the analysis for all non-Alu-exons and found that 7.8% of PTC+ transcripts were classified as NMD-sensitive. This is consistent with the previous finding that 9% of PTC+ transcripts were experimentally validated as NMD targets (*Lareau et al., 2007*), highlighting the difficulty of accurately predicting the recognition of PTCs by the NMD pathway. Given that ~40% of Alu-exon PTC+ isoforms are NMD-sensitive, Alu-exons are more efficient in eliciting NMD compared to other exons. This indicates that mechanisms may exist that protect other PTC+ exons from being recognised by the NMD pathway.

It was previously discovered that efficient NMD in mammalian cells requires the PTC to be at a distance of at least 55 nucleotides (nt) from the downstream exon-exon junction(s) in order to allow the exon junction complex (EJC) to recruit UPF1 (*Le Hir et al., 2000*; *Nagy and Maquat, 1998*; *Thermann et al., 1998*; *Kervestin and Jacobson, 2012*). To test whether the 55-nt rule explains the NMD-refractory PTC+ exons, we focussed on Alu-exons that were supported by junction-spanning reads on either side (798 exons) to ensure that the distance of the PTC to the next exon-exon junction is accurately defined. In line with the 55-nt rule, we found that in Alu-exons and all other types of alternative exons, PTCs are more frequently located less than 55 nt from the next exon-exon junction in NMD-refractory than in NMD-sensitive exons (*Figure 2F* and *Figure 2—figure supplement 1F*). Nevertheless, only one-third of the NMD-refractory Alu-exons can be explained by a short distance of the PTC to the next exon-exon junction. Moreover, most NMD-refractory Alu-exons with a PTC within less than 55 nt from the exon-exon junction still have additional junctions downstream (20 out of 29 Alu-exons), suggesting that EJCs deposited further downstream may not be able to activate NMD.

In summary, we found that Alu-exons are coordinately regulated by both splicing repression and NMD. While NMD accounts for the diminished expression of Alu-exon genes in hnRNPC-depleted cells, a substantial number of PTC+ Alu-exons is not cleared by the NMD pathway. Notably, two-third of these NMD-refractory Alu-exons cannot be explained by the 55-nt rule, suggesting that additional mechanisms might be involved in controlling NMD sensitivity.

## Intron-retaining transcripts are refractory to NMD and accumulate in the cytoplasm

Intrigued by the observation that many Alu-exons are NMD-refractory, we examined the splicing of Alu-exons in more detail. Using a modified DEXSeq approach to monitor intron events, we found retention of introns upstream or downstream of an Alu-exon for ~10% of Alu-exons in at least one of the tested conditions, including 40 cases of significant retention of both introns (adjusted p-value < 0.01; *Figure 3—figure supplement 1A*). All transcripts also include the Alu-exon sequence and are therefore referred to as 'intron-retaining Alu transcripts'. The Alu-exons in these intron-retaining Alu transcripts have weaker 5' and 3' splice sites (*Figure 3—figure supplement 1B*), indicating that intron retention might be a by-product of inefficient Alu exonisation. We did not observe intron in the majority of genes (*Figure 3—figure supplement 1C*) or a preference for retention of distal introns in Alu-exon genes compared to other genes (p-value = 0.2202, $\chi^2$ test). Compared to introns not containing an Alu-exon, introns flanking potential Alu-exons were three to four times more frequently retained in hnRNPC-depleted cells (*Figure 3—figure supplement 1D*). Hence, intron retention in hnRNPC-depleted cells is significantly associated with Alu-exons. We did not observed increased intron retention around repressed Alu-exons in UPF1-depleted samples (*Figure 3A*).

Past studies reported that transcripts with unspliced introns are retained in the nucleus (*Yap et al., 2012*; *Takemura et al., 2011*; *Taniguchi and Masuyama, 2007*; *Boutz et al., 2015*). However, we found that the intron-retaining Alu transcripts of *NUP133* and *MCM3* are readily detectable in cytoplasmic RNA (*Figure 3—figure supplement 2A*). Only the introns flanking the Alu-exon, but no other introns of the same transcripts are found in the cytoplasm (*Figure 3B* and *Figure 3—figure supplement 2B*). We did not observe accumulation of unspliced RNA in the nucleus for genes with intron-retaining Alu transcripts (*Figure 1—figure supplement 4D*), and thus, we found no evidence for reduced nuclear export of those transcripts. Intriguingly, the abundance of intron-retaining Alu transcripts was unchanged in UPF1-depleted cells, suggesting that they are

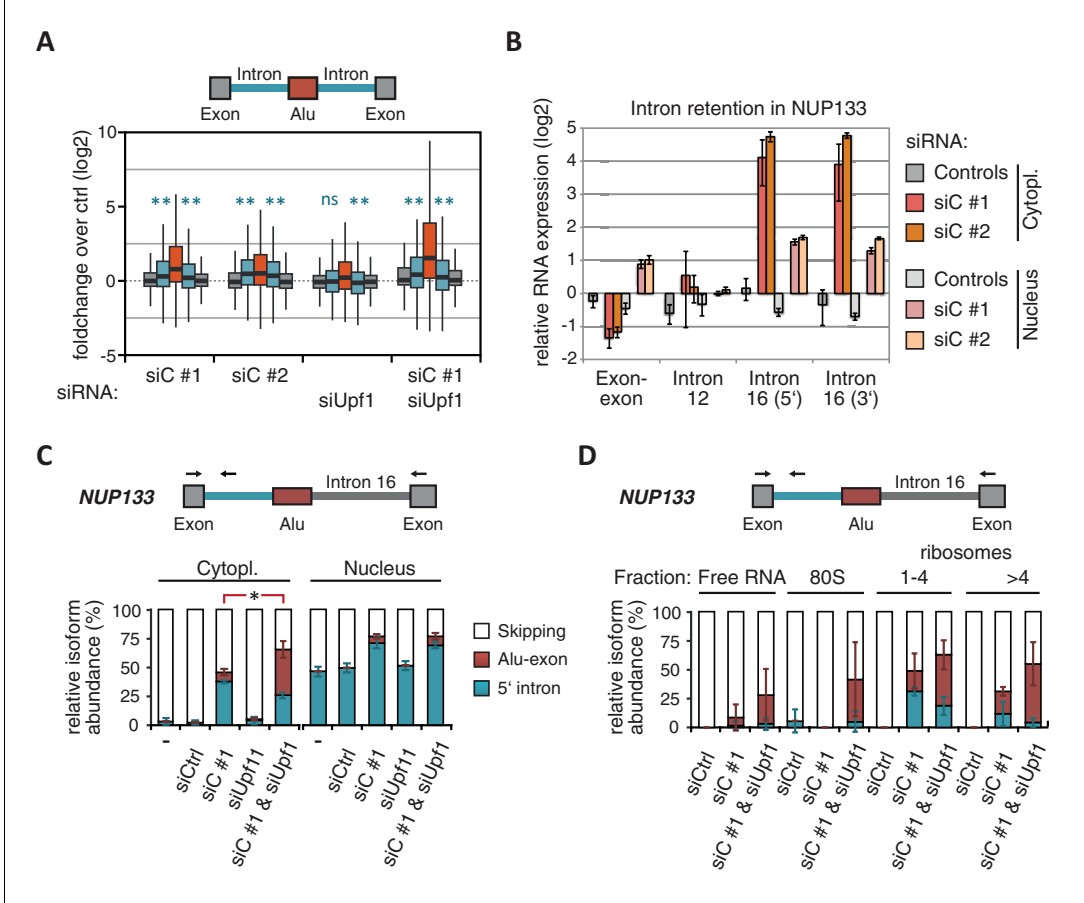

**Figure 3.** NMD-refractory transcripts are cytoplasmic and polysome-associated. (**A**) Boxplot presenting the fold changes (log$_2$) of Alu-exons as well as their two flanking introns and downstream and upstream exons upon hnRNPC or UPF1 depletion over control, as analysed by DEXSeq. Only significantly regulated Alu-exons from protein-coding genes were selected (n = 746, adjusted p-value < 0.01). To test for significant differences in intron retention, distribution of log$_2$ fold changes was tested against the null distribution with a two-sided Wilcoxon Rank Sum test. ** indicates p-value < 0.01. Note that the downstream intron in UPF1-depleted samples was significantly less often retained than in control, with median of -0.13 and 99% confidence interval (-0.20,–0.06). For hnRNPC-depleted samples, 99% confidence intervals confirmed a significant increase in retention. (**B**) Abundance of the introns flanking the Alu-exon in *NUP133* was measured by quantitative RT-PCR. As control, the overall mRNA abundance was quantified by an exon-exon-spanning primer, and abundance of the Alu-exon-free intron 12 was measured for comparison. Quantifications were performed for cytoplasmic and nuclear RNA of cells depleted of hnRNPC (siC #1 and #2), or control cells (no siRNA and control oligonucleotide) and normalised to the abundance of control mRNAs (*eIF4G* and *SDH*) in control cells (by ΔΔCt). Data are plotted as average of three independent replicates, error bars represent s.d.m. (**C**) The relative abundance of the intron-retaining Alu transcript of *NUP133* was measured in cytoplasmic and nuclear RNA samples as in (**B**), with abundance being measured relative to the Alu-exon transcript and the constitutively spliced transcript by semi-quantitative RT-PCR. To test for significance, one-way ANOVA was performed separately for cytoplasmic and nuclear samples. Multiple comparison correction was done according to Tukey's HSD. * indicates p-value < 0.05. Semi-quantitative RT-PCR analysis is averaged across three independent replicates, error bars represent s.d.m. (**D**) The relative abundance of the intron-retaining Alu transcript of *NUP133* was measured in RNA fractions from polysome gradients of cells depleted of hnRNPC (siC #1), or UPF1 and hnRNPC (siC + siUPF1), or control cells (no siRNA and control oligonucleotide). For details on fractions from polysome profile, see *Figure 3—figure supplement 2D*. Semi-quantitative RT-PCR analysis is averaged across three independent replicates, error bars represent s.d.m. *Figure 3—figure supplement 1* presents our analysis on retention of introns, which are flanking Alu-exons, and all other introns. *Figure 3—figure supplement 2* shows additional evidence that NMD-refractory transcripts are cytoplasmic and polysome-associated. *Figure 3—figure supplement 3* presents RNAseq traces as examples for intron retention in three genes, *NUP133*, *GMPS* and *C8ORF76*.

The following figure supplements are available for figure 3:

**Figure supplement 1.** Alu-exons with weak splice sites cause formation of intron transcripts.

**Figure supplement 2.** Cytoplasmic NMD-refractory transcripts.

*Figure 3 continued on next page*

*Figure 3 continued*

**Figure supplement 3.** Examples of intron-retaining Alu transcripts.

resistant to NMD (*Figure 3C* and *Figure 3—figure supplement 2C*). This is in spite of the fact that the corresponding Alu-exon transcripts, which include only the Alu-exon itself, are subject to NMD (*Figure 3C* and *Figure 3—figure supplement 2C*). Thus, the intron-retaining Alu transcripts appear to harbour certain features enabling them to evade NMD.

Besides the NMD-refractory Alu-exon transcripts, the intron-retaining Alu transcripts present a second group of transcripts in our study that are not efficiently cleared by NMD. Since PTC detection via NMD relies on translation (*Trcek et al., 2013*; *Singh et al., 2007*), translational inhibition could explain such NMD-refractory events. We therefore used polysome sedimentation experiments (polysome profiling) to test whether the NMD-refractory transcripts are associated with polysomes. To this end, we extracted untranslated ribosome-free RNA and 80S-associated RNA as well as RNA associated with 1–4 or more ribosomes (polysome fractions, *Figure 3—figure supplement 2D*). Semi-quantitative RT-PCR demonstrated that the intron-retaining Alu transcripts of *NUP133* as well as the NMD-refractory Alu-exon transcripts of *TNPO3* and *ZFX* are present in the polysome fractions, indicating that they might be translated (*Figure 3D* and *Figure 3—figure supplement 2E*). In summary, our data indicate that inclusion of intronic sequences flanking Alu-exons protects the resulting transcripts from NMD. It remains to be seen which elements are required for protection from NMD.

## Evolutionary progression of 3' splice sites is coupled with lengthening of the repressive U-tracts

It has been postulated that mutations introducing splice sites are the primary driving force in the emergence of Alu-exons (*Lev-Maor et al., 2003*; *Sorek et al., 2004*). Consistently, we found that exonising Alu elements have considerably stronger splice sites than other intronic Alu elements that do not give rise to Alu-exons (referred to as 'silent Alu elements'; *Figure 4A* and *Figure 4—figure supplement 2A*). Yet, exonising Alu elements have longer U-tracts than silent Alu elements (median length of 11 nt compared to 7 nt, *Figure 4A* and *Zarnack et al., 2013*), possibly suggesting a selection pressure for efficient hnRNPC repression. Previous phylogenetic maps across primate species showed that mutations in the splice site sequences usually occur significantly after integration of the Alu element (*Krull et al., 2005*; *Singer et al., 2004*), which would allow time for the U-tracts of the Alu-exon to decay through genomic drift. For example, the exonising Alu element in the *REL* gene (in the past called *c-rel-2*) obtained a 5' splice site only after the split of Old and New World monkeys (in the catarrhini lineage), even though the Alu element itself is present in both groups.

Alu-exons with shorter U-tracts are less repressed by hnRNPC (*Figure 4—figure supplement 1* and *Zarnack et al., 2013*). To understand how the interplay of positive and negative regulatory elements contributed to the formation of Alu-exons, we examined U-tract length and splice site strength in Alu-exons across the human as well as four primate genomes. First, we classified Alu-exons by sequence divergence (determined by the number of mutations from the Alu consensus sequence), which reflects the mutational load of an Alu element since its integration and is thus an approximate measure of evolutionary age (*Deininger and Batzer, 2002*; *Smit, 1999*; *dos Reis et al., 2016*). With increased sequence divergence, 3' splice site strength increases, while 5' splice site strength shows little variability (*Figure 4A*, and *Figure 4—figure supplement 2A*). In the oldest Alu-exons, the optimised 3' splice sites are accompanied by shorter U-tracts (*Figure 4A* and *Figure 4—figure supplement 2B*), which leads to reduced repression by hnRNPC (*Figure 4B*). In contrast, emerging Alu-exons do not seem to be under selective pressure to escape NMD, since around half of them introduce a PTC or frame-shift irrespective of the divergence of the underlying Alu element (*Figure 4—figure supplement 2C*). Together, this indicated that Alu-exons eventually escape strong hnRNPC repression by accumulating mutations that shorten their U-tracts.

To explore if evolutionary old Alu-exons transition towards *bona fide* exons, we made use of the GTEx V6p data (*GTEx Consortium, 2015*), and calculated percent exon inclusion (PSI) in various human tissues. Alu-exon age and 3' splice site strength weakly correlated with maximum inclusion

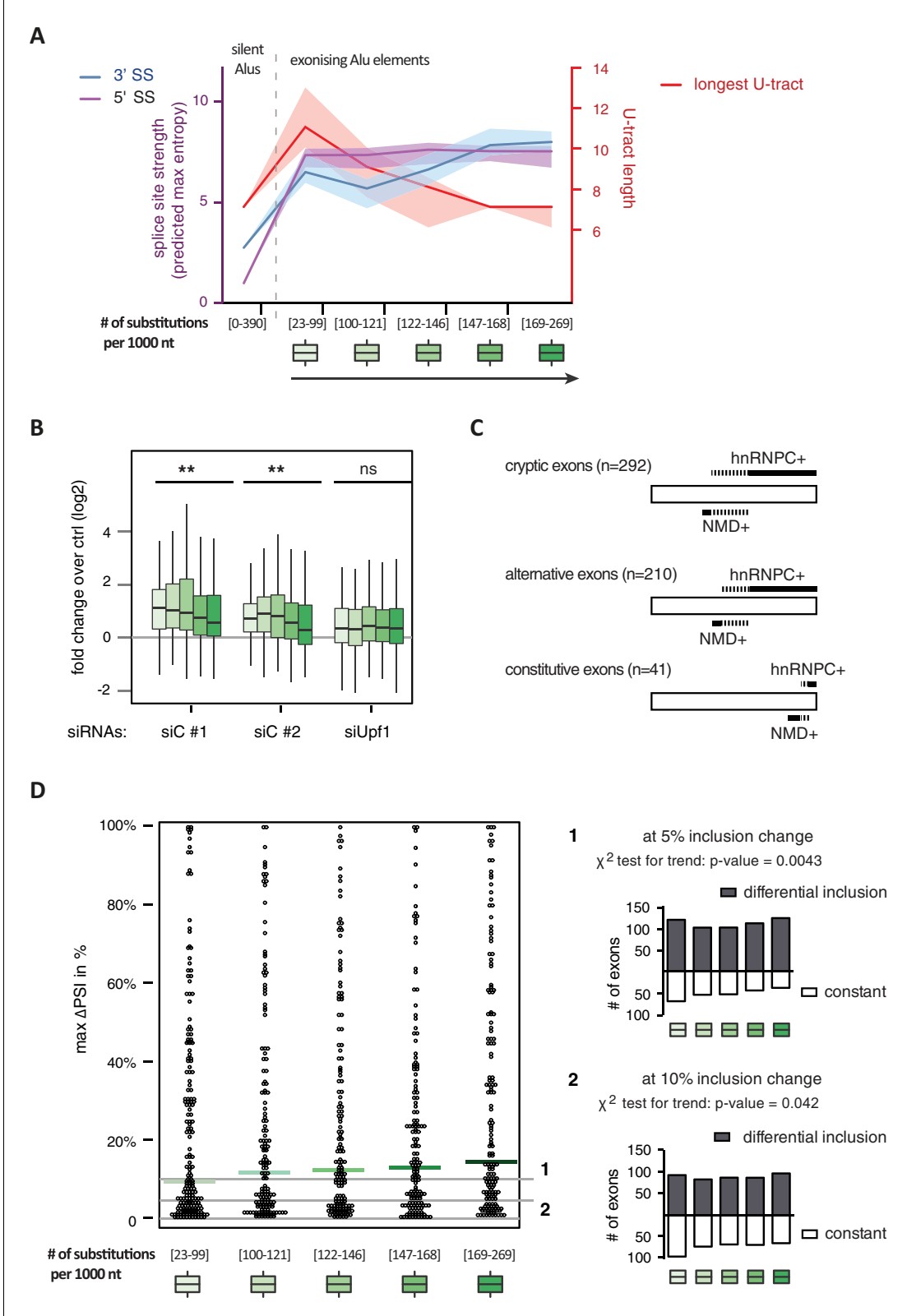

**Figure 4.** Alu-exon age correlates with loss of U-tracts and repression by hnRNPC. All 798 Alu-exons validated by junction-spanning reads at either splice site were stratified by the number of substitutions from the Alu consensus sequence provided by RepeatMasker (*Xiao et al., 2009*), as proxy for their evolutionary age, into five groups of roughly equal size (156–162 exons) with [23-99], [100-121], [122-146], [147–168] and [169–269] substitutions per 1000 nucleotides. (**A**) The median splice site strength and U-tract length of the different age groups of Alu-exons are shown on the left and right y-axis,

*Figure 4 continued on next page*

*Figure 4 continued*

respectively (purple, blue and red lines). 95% confidence intervals of the median were estimated by bootstrapping, and are plotted as area above and below the medians. The distributions of the splice site strengths and U-tract lengths in each group are shown in *Figure 4—figure supplement 2A and B*. For comparison, the median splice site strengths and U-tract lengths of all non-exonising ('silent') Alu elements are shown. (B) Changes in Alu-exon abundance of each group upon hnRNPC or UPF1 depletion as $\log_2$ fold changes (log2fc) over control, analysed by DEXseq. The correlation between substitutions in the Alu element and log2fc of each exon was tested by a linear model. ** indicates p-value < 0.01. (C) Depiction of the number of hnRNPC- and NMD-sensitive exons. Dashed lines visualise proportion of exons regulated by both hnRNPC and NMD. Only Alu-exons with an adjusted p-value < 0.01 are shown as hnRNPC+ or NMD+, Alu-exons with an adjusted p-value > 0.1 are assigned as non-regulated exons, and all other Alu-exons are ignored. (D) To test for differential splicing of Alu-exons across human tissues, we analysed GTEx expression data (*GTEx Consortium, 2015*) and calculated percent exon inclusion (PSI) values. For each exon with sufficient coverage by junction-spanning reads (n = 1039), the difference between the tissue with the lowest and highest inclusion is shown (max. ΔPSI), stratified by Alu-exon divergence. The median is shown as coloured line in each group. To illustrate the differences between Alu-exon groups, two arbitrary thresholds for 'differential inclusion' (5% and 10%) were used to set the number of exons that are differentially included or constant (right side). The substitution groups were then treated as contingency tables to test for differences in the number of differentially included exons with a $\chi^2$ test for trend. In *Figure 4—figure supplement 1*, we quantify the relationship between U-tract length and repression by hnRNPC on Alu-exons. *Figure 4—figure supplement 2* presents the complete distribution of the data on 3' and 5' splice site strength, U-tract lengths and $\log_2$ fold changes summarised in *Figure 4A–C*.

The following source data and figure supplements are available for figure 4:

**Source data 1.** Percent exon inclusion of Alu-exons in different human tissues.

**Figure supplement 1.** Repression of Alu-exons in dependence of U-tract length.

**Figure supplement 2.** Features of Alu-exons grouped by substitution rate.

---

levels (Spearman's correlation coefficient ρ = 0.09 and ρ = 0.1, both p-values < 0.05). Moreover, we also examined the maximum difference in exon inclusion across tissues to identify tissues-specific exons with a switch-like capacity (*Wang et al., 2008*). With increasing age, Alu-exons tend to be more frequently alternatively spliced between tissues (*Figure 4D*). Hence, we conclude that with increased age, Alu-exons tend to transition towards both higher and more tissue-specific inclusion.

As a second approach to study the paths of Alu-exon evolution, we examined the orthologues of exonising Alu elements in four primate species, including Old World and New World monkeys (chimpanzee, gibbon, rhesus macaque and marmoset). Since these split at different points from the human lineage, the presence of splice site sequences in other primates acts as an independent qualitative measure of evolutionary age. Based on our initial observation that 3' splice site optimisation is a crucial factor in Alu-exon evolution (*Figure 4A*), we classified the Alu-exons according to the evolutionary emergence of their 3' splice site. We considered 2876 Alu-exons that contain a 3' splice site with a score greater than three in human (predicted maximum entropy based on nucleotide sequence, *Yeo and Burge, 2004*), and for which the orthologous position could be found in at least one of the four primate species (*Figure 5—source data 1* and *Figure 5—figure supplement 1A*). We further classified these Alu-exons according to whether the cryptic 3' splice site is present in Old World and New World monkeys (based on the marmoset genome, 47% of exons), only in Old World monkeys (based on the rhesus macaque genome, 39% of exons), or only in the hominoidae lineage (gibbon or chimpanzee, 14%). The Alu-exons with a cryptic 3' splice site early in evolution had shorter U-tracts compared to those where it emerged later. Comparing substitution rates cross-confirmed that those were the most ancient Alu-exons (*Figure 5—figure supplement 1A*). The median U-tract length is 10 nt for Alu elements present in Old and New World monkeys (10–11 nt, 95% confidence interval [ci]), 11 nt for those specific for Old World monkeys (ci: 10–12 nt), 13 nt for those specific for hominidae (ci: 12–15 nt). Thus, the classifications based on either Alu element divergence in the human genome (*Figure 4A*) or on evolutionary emergence of a 3' splice site demonstrate that ancient Alu-exons have shorter U-tracts.

We considered an Alu-exon as evolutionary 'stable' throughout the examined evolutionary period if its 3' splice site was present in the most distant species to human (for example marmoset), and its strength was similar or stronger when compared to human (47.8% of Alu-exons). The remaining Alu-exons were further divided into those that lacked a 3' splice site in the most distant species ('emerging exons', 26.6%), and those with an existing 3' splice site that further increased in strength towards

the human lineage ('evolving exons', 25.6%). Strikingly, emerging exons have longer U-tracts as evolution proceeds towards the human lineage, coupled with their increasing 3' splice site strength, and the same trend is observed for evolving exons (*Figure 5A and B*). For instance, median U-tract length of evolving and emerging Alu elements gradually increases from 8 nt in marmoset to 10 nt and 12 nt in human, respectively (*Figure 5A*). The same coupling between increasing 3' splice site strength and U-tract length was also observed with the independent set of Alu elements that are unique to Old World monkeys (*Figure 5B*).

Overall, our intra- and inter-species analyses suggest that the U-tract is a crucial regulatory element in Alu elements, which is a target of selection during Alu exonisation, specifically at the stage of 3' splice site optimisation (*Figure 6A and B*). Long U-tracts couple the emerging 3' splice site with increasing splicing repression by hnRNPC, thus ensuring that most Alu-exons remain in a cryptic state. Yet, U-tract length and repression by hnRNPC decline with increasing evolutionary age of the Alu element, despite the remaining presence of the strong 3' splice site. This demonstrates that splicing repression by hnRNPC gradually decreases and predicts that Alu-exons escape splicing repression at a late stage of their evolution.

## Discussion

Intronic antisense Alu elements are prone to forming cryptic exons (*Sorek et al., 2002*). They present a major challenge to splicing fidelity but also open an opportunity for the emergence of new exons during evolution (*Ule, 2013*). Our study uncovers many new cryptic Alu-exons, raising the number of exonising Alu elements to over 6300. We demonstrate that splicing repression and NMD cooperatively control these exons to prevent the accumulation of potentially aberrant Alu-exon transcripts. U-tracts are longest in evolutionarily young Alu-exons, which leads to strong splicing repression by hnRNPC. Selection pressure for long U-tracts is strongest in Alu elements that show progressively stronger 3' splice sites when comparing orthologous Alu sequences of other primates to human. We speculate that splicing repression by hnRNPC serves as a molecular buffer during 3' splice site evolution at Alu elements, which allows the cryptic Alu-exons to undergo a stage of regulatory co-evolution before forming *bona fide* alternative exons. NMD targets Alu-exons to a similar extent irrespective of their evolutionary age, indicating that NMD is either tolerated or even beneficial to regulate the expression of Alu-exon transcripts. Notably, we find that NMD is inefficient in targeting intron-retaining Alu transcripts, highlighting the need for strong splicing repression to protect the transcriptome.

### Splicing repression and NMD cooperate to limit the abundance of Alu-exon transcripts

Depletion of hnRNPC triggers the exonisation of hundreds of Alu elements (*Zarnack et al., 2013*). Here, we show that depletion of UPF1 leads to a striking increase in the abundance of Alu-exons repressed by hnRNPC, demonstrating that Alu-exon transcripts are substrates to the NMD pathway. Since more than half of the Alu-exons are repressed by only one of the two pathways, we predict that splicing repression and NMD are needed to safeguard gene expression from Alu-exonisation.

Physiologically, NMD itself is subjected to regulation under a variety of conditions; for example, efficiency of NMD is decreased during differentiation of stem cells into neuronal precursors (*Lou et al., 2014*) and in response to stress conditions, such as hypoxia, nutrient deprivation and infection (*Fatscher et al., 2015*; *Hug et al., 2016*; *Gardner, 2008*). As a result, many transcripts that are normally degraded by NMD become upregulated under these conditions (*Gardner, 2008*; *Lou et al., 2016*). Since we observed that many Alu-exons promote NMD, we speculate that Alu-exons could have shaped the primate- and species-specific regulation of mRNA decay in response to variable efficiency of NMD. Potential PTCs remain relatively common in the highly divergent Alu-exons, and we observed that increased inclusion of Alu-exons often leads to NMD-dependent reduced expression levels. This suggests that most exonising Alu elements have not yet evolved to evade NMD. This could either be because selection pressure to remove PTCs is weak, or because the presence of the PTCs is of functional significance. While speculative, the later explanation has precedence: The Alu-derived alternative exon 10 in the gene *CD55* is NMD-sensitive (our study) and causes a frame-shift, thereby generating a secreted CD55 protein isoform that is required for regulation of the complement cascade (*Osuka et al., 2006*; *Caras et al., 1987*). This example illustrates a

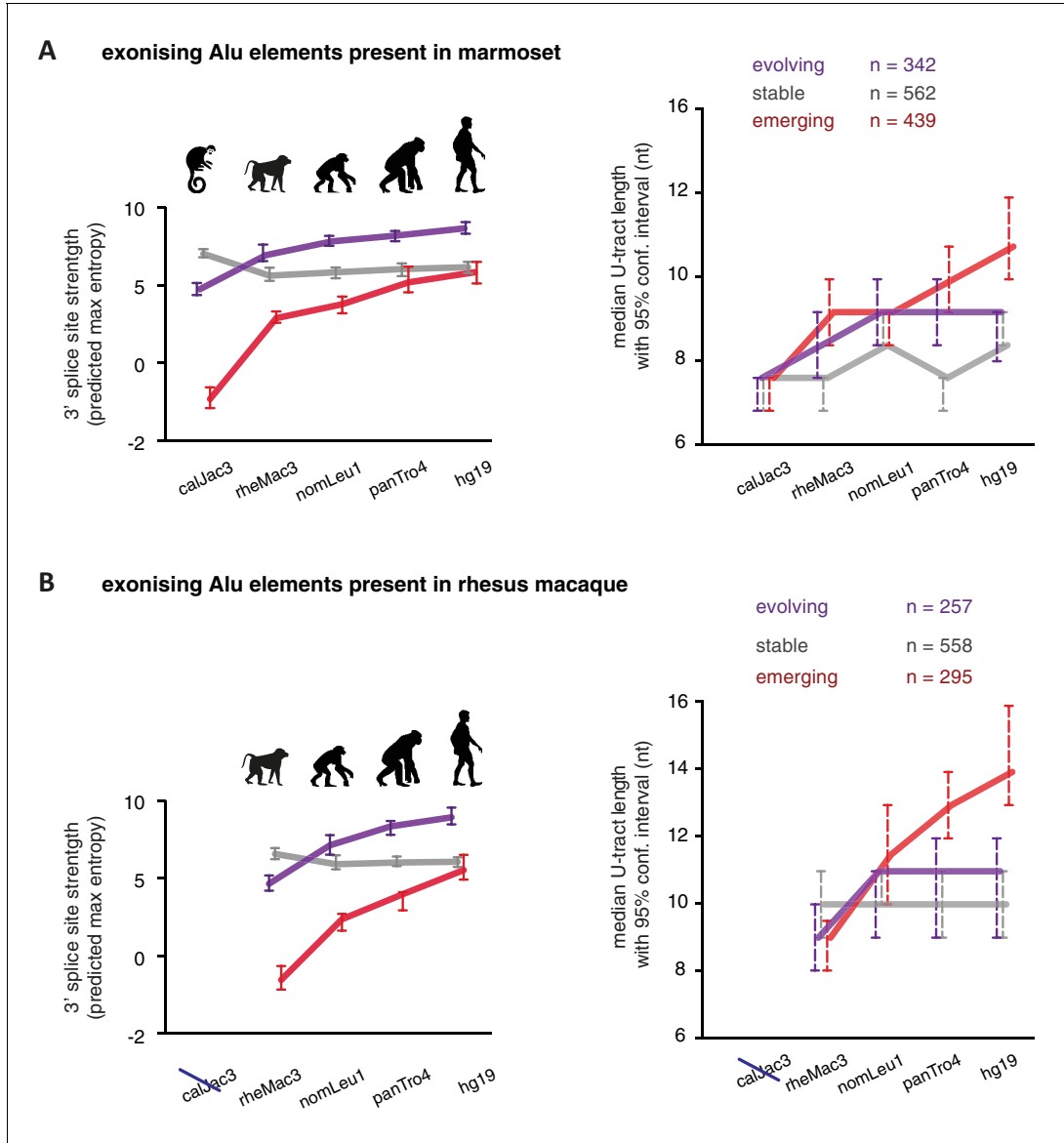

**Figure 5.** U-tracts of emerging Alu-exons lengthen in parallel to an abrupt gain of 3' splice site sequences. For all known Alu-exons found in this study or annotated in UCSC, orthologous regions were identified in four other primate genomes and scanned for the presence of an Alu element and a 3' splice site therein. We used the following genomes: *hg19* (human), *panTro4* (chimpanzee), *nomLeu1* (gibbon), *rheMac3* (rhesus macaque) and *calJac3* (marmoset). Alu-exons were split into three groups depending on the time point of 3' slice site emergence: Exons with an existing 3' splice site and little change in splice site strength across the five species were considered as 'stable', exons with an existing 3' splice site that gained strength towards the human lineage were considered as 'evolving', and exons in which the 3' splice site emerged after the split of Old and New World monkeys were considered as 'emerging'. (**A**) Alu-exons with orthologues present in marmoset (calJac3) were classified as described. Progression in predicted 3' splice site strength of the three groups of Alu-exons is shown on the left. Median lengths of the longest U-tract of the Alu elements in each species are shown on the right. Emerging Alu-exons were present in marmoset but only acquired a 3' splice site later in the evolutionary history of the primates. All data are presented as 95% confidence intervals of the median estimated by bootstrapping. (**B**) Alu-exons with orthologues present in rhesus macaque (rheMac3) but not in marmoset were classified as described. Predicted 3' splice site strength and median U-tract length in each species as in (**A**). Emerging Alu-exons were not present in marmoset but in rhesus macaque, and acquired a 3' splice site later in the evolutionary history of primates. All data are presented as 95% confidence intervals of the median estimated by bootstrapping. *Figure 5—figure supplement 1* presents the complete distribution of U-tract lengths summarised in *Figure 5A and B*, as well as the data on the evolutionary youngest Alu-exons specific to the hominoidae lineage.

The following source data and figure supplement are available for figure 5:

**Source data 1.** List of Alu-exons across our datasets and UCSC annotation, including cross-species annotation.

*Figure 5 continued on next page*

*Figure 5 continued*

**Figure supplement 1.** U-tract length of Alu-exons traced throughout primate orthologues.

scenario that might be common for established Alu-exons: they provide an opportunity to evolve novel protein sequences, but at the cost of reduced gene expression due to a residual sensitivity to NMD.

## NMD is inefficient for many PTC-containing transcripts, particularly intron-retaining Alu transcripts

Even though the vast majority of Alu-containing transcripts contain PTCs, only about 40% are efficiently degraded by NMD. Yet, only ~8% of annotated exons with a PTC undergo detectable NMD (our study and *McIlwain et al., 2010*), suggesting that other exons with PTCs are more efficient in evading NMD than cryptic Alu-exons. Our analyses confirm previous findings that a PTC further than 55 nt from a downstream EJC is not sufficient to reliably predict NMD substrates (*McIlwain et al., 2010*; *Lindeboom et al., 2016*). A major finding of our study are over 300 transcripts, in which intronic sequences flanking an Alu-exon are retained. These intron-retaining Alu transcripts evade recognition by NMD despite the fact that they accumulate in the cytoplasm. This is unexpected since past studies found that intron retention usually triggers either nuclear retention or NMD, separate mechanisms that are also exploited for specific regulation, e.g. to downregulate genes during cellular differentiation (*Yap et al., 2012*; *Braunschweig et al., 2014*; *Wong et al., 2013*). It is even more surprising since the corresponding Alu-exon transcripts are efficiently degraded by NMD, even though the intron-retaining Alu transcripts often contain more PTCs in their open reading frame than the Alu-exon transcripts. We observed that NMD-resistant Alu-exon transcripts of *TNPO3* and *ZFX* as well as the intron-retaining Alu transcript of *NUP133* are associated with polysomes in a similar ratio as the conventionally spliced transcripts. Yet, more tests will be needed to address whether the same applies to the multitude of intron transcripts observed in cells depleted of hnRNPC. Moreover, it will be exciting to see whether the polysome-associated intron transcripts are indeed being translated, for instance by mining deep proteomics data collected from hnRNPC-depleted cells.

Several scenarios beyond lack of translation could explain the NMD resistance of intron-retaining Alu transcripts. Firstly, the retained intronic sequences are usually longer than regular exons, likely resulting in an increased distance between any PTC and the next EJC. Secondly, splicing factors could remain bound on the retained introns and may prevent EJC assembly. Thirdly, it is tempting to speculate that the retained introns may contain NMD-inhibitory sequences. A recent study described NMD-inhibitory, AU-rich sequences in some 3' UTRs, which are sufficient to block NMD in heterologous reporters (*Toma et al., 2015*). Since introns generally show a higher A/U content than exonic sequences or Alu elements (*Amit et al., 2012*; *Louie et al., 2003*), AU-rich or other sequence elements in the intron-retaining Alu transcripts may allow them to evade NMD. Finally, we can not formally exclude that intron-retaining Alu transcripts are sequestered and at the same time degraded in the nucleus, which could partly contribute to their reduced gene expression. Further studies of NMD-inhibitory sequences may help to understand why so many PTC+ transcripts evade NMD.

## Evolutionary coupling of positive and negative splicing elements

Our intra- and inter-species analyses reveal longest U-tracts at Alu elements that recently acquired a 3' splice site, suggesting strongest selection pressure for repressive U-tracts during the initial evolutionary phase of cryptic Alu-exons (summarised in *Figure 6A*). Interestingly, Alu-exons with stable 3' splice site across primate species show an average U-tract length of 8–10 uridines (*Figure 5A and B*), which is at the lower limit of 9 to 10 uridines necessary to allow cooperative binding by the two RRMs of hnRNPC (*Cieniková et al., 2015*). We propose that a cryptic 3' splice site is best tolerated within an Alu element that already contains a long U-tract, while emergence of the same site in the absence of a long U-tract is subject to negative selection. Consistent with this hypothesis, it was proposed that low inclusion of newly emerging exons is required to reduce negative selection against them (*Xing and Lee, 2006*). However, since only a limited number of primate genomes are analysed,

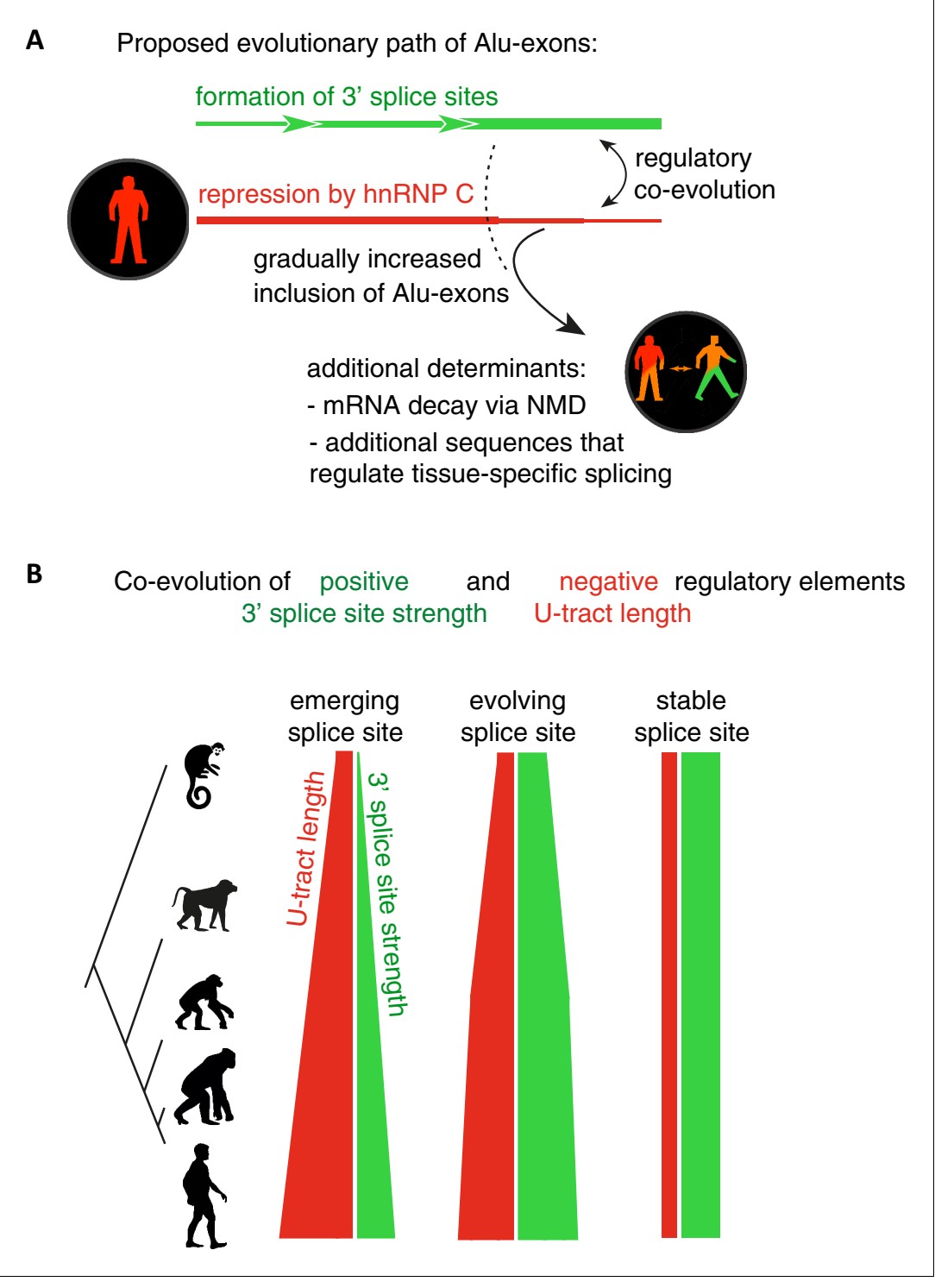

**Figure 6.** Repressive elements co-evolve with splice site sequences at cryptic exons. Alignment within primate genomes of Alu elements that contain human Alu-exon reveals a tight coupling between repressive U-tracts and the variation of 3' splice site strength. (**A**) As the strength of 3' splice sites increases, the repressive U-tracts are lengthened, which recruits hnRNPC to prevent splicing of Alu-exons. Hence, the splice site sequence and the repressive U-tract undergo an evolutionary dynamic that we refer to as regulatory co-evolution. Splice site sequences without a nearby repressive element are depleted, likely due to strong negative selection. Selection pressure for long U-tracts decreased for as-yet unknown reasons at the more ancient Alu-exons (those that contain a splice site in a distant primate species, or their sequence diverges from the Alu consensus), and these exons are

*Figure 6 continued on next page*

*Figure 6 continued*

less repressed by hnRNPC and have an increased incidence of tissue-specific splicing. At this stage, abundance of the Alu-exon isoform is determined also by its ability to trigger NMD, and likely other factors such as tissue-specific splicing factors. (B) Based on the variation of the 3' splice sites between primate species, we characterised three evolutionary groups of Alu-exons. Most human Alu-exons have stronger 3' splice sites in human compared to New World monkeys, and these exons also have longer repressive U-tracts in human. These exons are split according to the evolutionary trajectory of their 3' splice site sequence into emerging, evolving or stable 3' splice sites. The most ancient Alu-exons, which have a strong and stable 3' splice site, lack any trend towards longer U-tracts. This demonstrates that tight coupling between positive and negative splicing elements establishes a balanced regulatory environment at the newly emerging exons.

we might miss short periods of rapid evolution. It remains possible that an unrepressed strong 3' splice site can be tolerated within an Alu element but is commonly followed by a period of U-tract lengthening due to strong positive selection against inclusion of the Alu-exon. While we cannot fully distinguish between scenarios of negative and positive selection, it is clear that long U-tracts are selected as balancing elements at splice sites increasing in strength due to random genomic drift. Repression by the hnRNPC:U-tract interaction prevents abrupt increases in Alu-exon inclusion at strengthening 3' splice sites, which could otherwise disrupt gene expression due to NMD or produce toxic proteins.

The selection pressure for long U-tracts favours Alu-exons with repressive elements in proximity to positive elements; effectively, positive and negative splice sequences co-evolve (*Figure 6B*). Notably, this 'regulatory co-evolution' takes place while the Alu-exons are still in a cryptic state, before they acquire any functions. Regulatory co-evolution may act to ensure a more gradual evolutionary path at emerging exons in general, and represent a crucial step in the emergence of new splice site sequences. Co-evolution of positive and negative splice elements has already been described at a set of constitutively included exons (*Xiao et al., 2009*). Burge and colleagues proposed that intronic G-tracts attract hnRNPH at 5' splice sites to enhance exons with intermediate-strength 5' splice sites, and thus act as 'genetic buffers' at constitutive exons with mutations that weaken the 5' splice site sequence. This illustrates that regulatory co-evolution describes a dynamic process in which evolving splice sites sequences are coupled to regulatory RBP binding sites, in order to maintain a stable level of exon recognition. This process can thus either maintain exons in a cryptic state (in the case of U-tracts at Alu-exons), or in a constitutive state (in the case of G-tracts).

## Does regulatory co-evolution ensure the gradual emergence of tissue-specific exons?

If co-evolution of positive and negative splicing elements ensures that emerging Alu-exons remain in a cryptic state, how does an exon progress from a cryptic to a *bona fide* exon? We propose that regulatory co-evolution opens an opportunity for evolutionary tinkering of the exon's sequence (*Feschotte, 2008*; *Jacob, 1977*; *Sibley et al., 2016*). While repressed through its U-tract, a cryptic exon might acquire mutations that decrease its toxicity, thus gradually decreasing the selection pressure for the repressive elements. However, it might be far more likely that the regulatory elements diversify. If new sequence elements recruit repressors other than hnRNPC, which are not ubiquitously expressed, they could establish tissue-specific regulation of individual exons. This would also reduce negative selection, since cell-type-specific expression will mitigate the impact of a single exon on a multi-cellular organism. We favour this scenario, since we observed both an increase in maximum inclusion of an Alu-exon with increased Alu-exon age, and a trend towards a switch-like inclusion in individual tissues. We were not able to identify other splicing regulators competing with hnRNPC by iCLIP binding patterns (data not shown, and *Kelley et al., 2014*). Hence, such a scenario remains hypothetical, since no tissue-specific splicing repressor is yet known to regulate inclusion of Alu-exons in a pronounced manner. Since it is likely that each tissue-specific Alu-exon is regulated by a unique combination of RBPs, it will require further studies to fully unravel these more complex regulatory networks.

## Implications for disease-causing mutations

Our model of regulatory co-evolution of splice site sequences and RBP binding sites suggests that this process allows cells to maintain a delicate balance at newly emerging exons. Cryptic Alu-exons with strong splice sites are coupled to repressive binding sites of hnRNPC, which allows them to be tolerated and thus to accumulate within primate transcripts. However, the large number of these cryptic Alu-exons exposes them to germline or somatic mutations that could disrupt the repressive U-tracts, thus leading to inclusion of potentially deleterious exons. We therefore propose that the cryptic Alu-exons reported in our study could be considered for interpretation of whole-genome sequencing studies that aim to discover new disease-causing mutations. Unravelling the coupling of negative and positive splicing elements at cryptic exons would thus help to understand how mutations within the vast intronic regions of our genes cause disease.

# Materials and methods

## Cell culture and siRNA transfection

HR1 cells (*Ewings et al., 2007*) were a gift from Simon Cook (Babraham Institute, Cambridge) and are a stable derivative from HEK293 cells (ATCC Cat# CRL-1573, RRID:CVCL_0045). HR1 cells were maintained in DMEM with 10% FBS and 400 µg/ml G418 at 37°C with 5% $CO_2$ injection, and routinely passaged twice a week. Cells were regularly cultured for 3 days in antibiotic-free medium and tested for mycoplasma using either the LookOut Mycoplasma PCR Detection Kit or the MycoAlert Mycoplasma Detection Kit (Lonza).

To deliver siRNAs, Lipofectamin RNAiMax (Life Technologies, now Thermo Fisher Scientific, Waltham, MA, USA) was used according to manufacturer's recommendations. HR1 cells were reverse-transfected with 5 nM siRNA targeting *HNRNPC* mRNA (HSS179304 and HSS179305) or *UPF1* mRNA (HSS109172) as well as a siRNA negative control with medium GC content (Invitrogen, Cat. number 12935–300, now Thermo Fisher Scientific). Depletion of hnRNPC was achieved 48 hr after initial siRNA transfection. For depletion of UPF1, cells were transfected a second time after 48 hr, and collected 96 hr after the first siRNA transfection.

## Nucleo-cytoplasmic fractionation

Cells were washed with ice-cold PBS and lysed with NP40E-CSK buffer (350 µl per well of a 6-well plate or 600 µl per 10 cm² dish). NP40E-CSK buffer was similar to the cytoskeleton buffer used in (*Reyes et al., 1997*) and composed of 50 mM Tris-HCl (pH 6.5), 100 mM NaCl, 300 mM sucrose, 3 mM $MgCl_2$, 0.15% NP40 and 40 mM EDTA. Lysis was allowed to proceed for 5 min on ice. Cytoplasmic supernatant and pelleted nuclei were separated at 4°C, 5000 x g for 3 min. The cytoplasmic supernatant was cleared with another spin at 4°C, 5000 x g for 3 min and another spin at 4°C, 10000 x g for 10 min. Nuclei were washed with 400 µl NP40E-CSK buffer and incubated for 5 min under rotation to ensure complete cell lysis. After repeat of the centrifugation step, nuclei were lysed in 300 µl RIPA lysis buffer (50 mM Tris-HCl [pH 7.4], 100 mM NaCl, 1% Igepal CA-630 [Sigma I8896, St. Louis, MO, USA], 0.1% SDS, 0.5% sodium deoxycholate) and sonicated, either 2 x 3 s pulses on a one-sample device or 5 x 30 s pulses in a BioRuptor water bath device. For preparation of RNA for RNAseq, an additional wash step with 180 µl NP40E-CSK buffer was done before nuclei rupture. Subsequently, RNA was isolated using Trizol LS (Invitrogen) according to manufacturer's recommendations.

## Quantitative and semi-quantitative RT-PCRs

Reverse transcription was done with 500 ng of RNA using RevertAid enzyme (Fermentas, now Thermo Fisher Scientific) according to manufacturer's recommendations. The reverse transcription was primed with equal parts of random N6 and N15 oligonucleotides (Sigma) at 100 µM final concentration. For quantitative RT-PCR, the amplification of the cDNA of interest was normalised by the delta-delta Ct method to the geometrical mean of house keeping genes measured in parallel in all assays (*EIF4G* and *SDH*). All quantitative RT-PCR assays were done in technical replicates and averaged before data analysis. For semi-quantitative PCR, we ran 35 cycles of amplification with the primers indicated in each figure, and quantified the abundance of each product using Qiaxcel (Qiagen,

Hilden, Germany) gel electrophoresis. Primer sequences and details are provided in *Supplementary file 1*.

## Generation of RNAseq libraries and high-throughput sequencing

To analyse RNA export in the absence of hnRNPC, cells transfected with siRNAs targeting hnRNPC or control siRNAs were induced with 100 nM 4-hydroxytamoxifen, collected at three time points (0, 30 and 60 min) and subjected to subcellular fractionation to obtain cytoplasmic and nuclear RNA. The experiment was designed as three-factor block design with in total 24 samples: RNA fractions were collected across three time points, with two control siRNAs and two siRNAs against hnRNPC. We used the TruSeq Unstranded RNAseq Library kit (Illumina, San Diego, CA, USA) to generate unstranded RNAseq libraries according to manufacturer's recommendations (used in *Figure 1*). To analyse Alu-exon inclusion in absence of hnRNPC and UPF1, biological triplicates were collected for each condition and in total 15 stranded RNAseq libraries were generated using the TruSeq mRNA Library kit (Illumina) according to manufacturer's recommendations (used in *Figures 2–4*).

All libraries were sequenced on Illumina HiSeq2 machines in a single-end manner with a read length of 50 nt (unstranded) or 100 nt (stranded). Before mapping the reads, we removed adapter sequences by using the FASTX tool kit version 0.7, discarded reads shorter than 35 nt and in case of the stranded RNAseq data, trimmed all reads to 75 nt length. RNAseq reads were mapped to UCSC hg19/GRCh37 genome assembly using TopHat v2.0.5, RRID:SCR_013035 (*Kim et al., 2013*), allowing up to two mismatches. RNAseq data files of rRNA depleted cytoplasmic and nuclear RNA from cells depleted of hnRNPC combined with a time-course of RAF1 kinase induction are deposited on EBI ArrayExpress under the accession number E-MTAB-4009. RNAseq data files of polyA-selected RNA from cells depleted of hnRNPC and UPF1 are deposited under E-MTAB-4008.

## De novo identification of cryptic exons and Alu-exons

In order to predict exons from our RNAseq data, we ran Cufflinks (version 0.9.3, -min-isoform-fraction 0, *Trapnell et al., 2012*, RRID:SCR_013307) on the collapsed reads from all samples of our stranded RNAseq data and then extracted the exons of all predicted transcripts. Cufflinks predicted a total of 387,009 exons. All exons with one or both splice sites residing within an antisense Alu element (as annotated by RepeatMasker, *Smit et al., 1996-2010*) were assigned as Alu-exons. In order to minimise noise, we kept only exons that were predicted as part of multi-exon transcripts and were supported by at least one junction-spanning read. To ensure the best possible annotation of Alu-exons, we included exons which were previously identified in HeLa cells following the same annotation strategy (*Zarnack et al., 2013*). This identified 1595 additional Alu-exons in the HR1 data which otherwise did not have a supporting junction-spanning read. Altogether, we generated a list of 5202 Alu-exons from RNAseq data. We found that UCSC annotation contained another 1107 exons originating from antisense Alu elements, placing the total number of antisense Alu elements known to be on the path of exonisation at 6309. For all analyses in this study, we ignored exons originating from Alu elements in sense orientation.

All exons that were not annotated in UCSC gene annotation (hg19) were referred to as 'cryptic Alu-exons'. In addition, the annotated Alu-exons were further classified into alternative and constitutive Alu-exons based on UCSC annotation of alternative exons ('UCSC Alt Events') and RefSeq exon annotations: Alu-exons not annotated as 'alternative' in UCSC and part of RefSeq transcripts were classified as constitutive, all others were treated as alternative Alu-exons.

## Analysis of differential gene expression and differential exon inclusion

Analyses of differential gene expression were performed using DESeq and DESeq2, RRID:SCR_000154, (*Anders and Huber, 2010*; *Love et al., 2014*) with gene coordinates based on ENSEMBL annotation (version 72). To combine the results from both siRNAs targeting *HNRNPC*, we used a conditional thresholding approach, calling expression changes as significant if they had an adjusted p-value < 0.01 in at least one of the two hnRNPC depletion conditions and an adjusted p-value < 0.05 in the other.

Differential splicing was determined using DEXSeq, RRID:SCR_012823 (*Anders et al., 2012*). The two hnRNPC depletion conditions were integrated by conditional thresholding as described above. To monitor the differential abundance of introns due to intron events, we extracted intron

coordinates based on ENSEMBL exon annotations (version 72): For all Alu-exons in the RNAseq dataset, we identified the annotated upstream and downstream exons and extracted the intervening introns. We ignored introns of less than 60 nt in length and shrank all remaining introns by 25 nt on either side to avoid reads contributing to intronic coverage from exons, which use suboptimal close-by splice sites in addition to the annotated splice sites. This resulted in a total of 3244 introns with sufficient coverage for our DEXSeq analysis. 409 of 3244 tested Alu-exon flanking introns showed significant retention under hnRNPC and UPF1 co-depletion (adjusted p-value < 0.01). Of those, 204 are flanking the 746 significantly regulated Alu-exons.

## Analysis of exons regulated by hnRNPC and NMD

To group exons according to their susceptibility for hnRNPC or NMD, we compiled a list of exons that were significantly regulated upon hnRNPC and/or UPF1 depletion according to DEXSeq (adjusted p-value < 0.01, 12,323 exons including 746 Alu-exons). For each exon in each sample, we used the number of reads on a single exon divided by the sum of reads on all exons of the same gene as an estimate of exon inclusion (EI).

EI = # of exon reads / # of reads on all exons of the same gene.

The estimate of exon inclusion was averaged across biological triplicates of each condition. Since we only selected significantly regulated exons in step 1, we were confident that the variation between replicates of this set of exons is small compared to the effect size of the knockdowns. We then calculated the change in inclusion upon depletion of hnRNPC, UPF1 or both compared to control:

ΔEI = EI (depletion) / EI (control)

To be counted as an hnRNPC-sensitive exon, the average change in inclusion in hnRNPC-depleted samples had to account for at least 40% of the change in the hnRNPC/UPF1 co-depleted samples:

ΔEI (mean siC #1/siC #2) / ΔEI (siC #1+siUPF1) > 0.4

All other events were rated as hnRNPC-refractory (hnRNPC-). To be counted as a NMD-sensitive exon, the change in inclusion in UPF1-depleted samples had to account for at least 10% of the change in the hnRNPC/UPF1 co-depleted samples:

ΔEI (siUPF1) / ΔEI (siC #1+siUPF1) > 0.1.

If ΔEI (siUPF1) / ΔEI (siC #1+siUPF) was less than 0.05, the exon was called NMD-refractory. All other exons were classified as 'unassigned'.

Non-regulated Alu-exons (NMD-, hnRNPC-) were Alu-exons expressed in control samples (average of at least five reads in control samples) and not significantly affected by any of the knockdown conditions (adjusted p-value > 0.1).

## Analysis of frame-shifts and premature stop codons in regulated exons

We selected the longest annotated transcript for each protein-coding gene from UCSC annotation including CDS coordinates and identified the position of the exon-of-interest within the transcript using the *intersect* function from bedtools (v2.22.1). Next, we determined the length of exonic sequence from the annotated start codon to the start of the exon-of-interest to determine the frame of the exon and predicted stop codons within that frame. If an exon did not contain a PTC in frame, we calculated if its length is a multiple of 3, and annotated exons as 'PTC'- or as 'potential frame-shift'. For PTC+ exons, we calculated the distance of the PTC to the next exon-exon junction.

To ensure sufficient accuracy of our frame prediction, we only used exons for which both junctions were supported by junction-spanning reads in our RNAseq datasets. As expected, exons annotated as constitutive were devoid of PTCs, while ~15% of annotated alternative exons contained a PTC.

## Analysis of splice site strength and U-tract length of Alu elements and Alu-exons

Splice site strengths were predicted based on maximum entropy with MaxEntScan (*Yeo and Burge, 2004*). For Alu-exons, we used UCSC annotation or Cufflinks prediction to identify 3' and 5' splice sites. For intronic antisense Alu elements (n = 331,451), we predicted the strength of the strongest splice site in the following manner: For 5' splice sites, we searched for ACAGG sequences, corresponding to the conserved consensus sequence fragment giving rise to 5' splice sites within Alu

elements (*Sorek et al., 2004*). The 5' splice site sequence itself is anchored at the CAG, corresponding to ACAG|GNNNNN. For 3' splice sites, we searched for TGAG|AnGG and GAGAnAG| (the expected 3' splice site is marked by AG|), which correspond to the most frequently used 3' splice site sequences within Alu elements, using either the proximal or the distal AG (*Lev-Maor et al., 2003*). For Alu elements with more than one putative splice site, we used the strongest. In total, we found potential 5' splice sites in 285,924 Alu elements and potential 3' splice sites in 166,353 Alu elements.

To find the longest U-tract of each element, we compiled a list of sequences for all antisense Alu elements, using always the last 350 nt from the end of the Alu element. We then identified all continuous U-tracts and selected the longest U-tract within each element. Only U-tracts of a length of at least 4 nt were considered, elements with a shorter U-tract were counted as 3-nt U-tracts (6885 Alu elements).

## Analysis of intron retention events

To identify intron retention events, we created two custom annotations. First, we included all introns flanking Alu-exons in our custom annotation. Introns were cut by 25 nt on either side to avoid counting reads from alternative splice sites of an Alu-exon, and we removed introns that contained cryptic exons (predicted by Cufflinks, see above). Then, we ran DEXseq to identify introns retained in hnRNPC- or UPF1-depleted cells or after hnRNPC and UPF1 co-depletion. Second, we created a custom annotation in the same manner, but including all introns defined as the gaps between the exons of a single gene (including our annotation of cryptic Alu-exons assigned to their host gene). Next, we removed all introns that overlapped with any annotated exon of another gene. This annotation covered 274,082 exons and 209,640 introns, including 6975 introns flanking an Alu-exon. Again, we run DEXseq to identify candidate introns with significant retention. Since this second approach is highly inflated by introns, all analysis on intron-retaining Alu transcripts refers to the first approach with exception of our efforts to estimate how frequently other introns are retained in comparison to introns flanking Alu-exons (*Figure 3—figure supplement 1C and D*).

## Analysis of differential exon inclusion in human tissues

To analyse differential inclusion of Alu-exons across human tissues, we retrieved data on mapped junctions from the V6p release of the GTEx consortium (http://www.gtexportal.org/home/, Ref *GTEx Consortium, 2015*). We identified junction-spanning reads to Alu-exons in a 5 nt grace window around the splice sites of all Alu-exons and used those to identify the 5' and 3' splice site of the upstream and downstream exon. We then calculated percent exon inclusion (PSI) as the ratio of inclusion junction reads (average of up+downstream junctions) to total junction reads (average of up+downstream junctions + skipping junctions), and inclusion within each tissue as average of all samples. To ensure sequencing depth and gene expression were sufficient to calculate exon inclusion, we only used exons with at least 200 reads across the 8555 samples (average of up+downstream junctions or skipping junctions). We also restricted the analysis to exons from protein-coding genes. In total, we covered 1139 Alu-exons across 30 tissues, which were adipose tissue, adrenal glands, bladder, blood, blood vessels, brain, breast, cervix/uterus, colon, esophagus, fallopian tube, heart, kidney, liver, lung, muscle, nerve tissue, ovary, pancreas, pituitary, prostate, salivary glands, small intestine, spleen, skin, stomach, thyroid, testis, uterus and vagina.

## Classification of Alu elements by divergence or evolutionary dynamics, and analysis of their 3' splice sites and U-tracts

To classify Alu elements by the divergence of their sequence in human genome compared to the Alu consensus, we used the nucleotide difference per 1000 nt, which is provided for each Alu element in the human genome by the RepeatMasker table (*Smit et al., 1996-2010*, hg19, Repeat Library 20090604). Analysis of sequence divergence is shown for Alu-exons that were supported by junction-spanning reads on both sides in our RNAseq data (798 exons), which accurately identified the splice site positions. To classify Alu elements by their evolutionary dynamics, we started with the complete set of 6309 non-overlapping Alu-exons but only examined human Alu-exons that had a 3' splice site sequence (3' SS) with a predicted strength higher than 3, to narrow down to Alu-exons which have acquired a 3' splice site sequence in human. We then used the UCSC Genome Browser

LiftOver tool (*Rosenbloom et al., 2015*) to obtain orthologous genomic loci of the 3' splice site of human Alu elements in the representative species for New World monkeys (marmoset, calJac3), Old World monkeys (rhesus macaque, rheMac3) and hominidae (gibbon, nomLeu1 and chimpanzee, panTro4) lineages. We first classified each Alu element based on the most distant species to which we could lift over its genome coordinates. We then collected the sequence 20 nt upstream to 3 nt downstream of the 3' SS for each species, and examined the maximum entropy of its sequence with MaxEntScan (*Yeo and Burge, 2004*).

Next, we classified the Alu elements based on the evolutionary dynamics of their 3' SS. 'Emerging' Alu-exons have a 3' SS with a score less than three in the most distant species. 'Stable' Alu-exons have a 3' SS higher than three in the species most distant to human, and its strength increased towards human by less than 1. 'Evolving' Alu-exons have a 3' SS higher than three in the species most distant to human, and its strength in human increased by more than 1. For example, if the score in marmoset is 2.5 and 4.5 in human, then the Alu exon is considered as 'emerging', if it is four in marmoset and 4.5 in human, then it's considered as 'stable', and if it is four in marmoset and six in human, then it i s considered as 'evolving'.

To determine the longest U-tract within the Alu element in each species, we first determined the distance D between the start of the antisense Alu element and the 3' SS in human genome. We then defined the region for analysis of U-tract as the window from [D+20 nt] upstream to [200 nt-D] downstream of the 3' SS.

Since annotation of Alu elements in non-human genomes is often incomplete, especially in the region of the first U-tract, our analysis ensured that the region analysed was broad enough to identify the full sequence of the longest U-tract derived from the Alu element. Confidence intervals of U-tract length and 3' splice site strength were estimated by bootstrapping (see below).

## Analysis of publicly available datasets

The Illumina BodyMap 2.0 data were downloaded from ArrayExpress (E-MTAB-513). Before mapping the reads, we removed adapter sequences by using FASTX-Toolkit version 0.7. RNAseq reads were mapped to the UCSC hg19/GRCh37 genome assembly using TopHat v2.0.5 (*Kim et al., 2013*), allowing up to two mismatches. We used the paired-end sequencing data; however, single-end sequencing data gave comparable results (data not shown).

To analyse the expression of Alu-exon containing genes, we focused on protein-coding genes and assigned the Alu-exons as follows: The longest annotated transcript was retrieved from ENSEMBL v72 to define the gene coordinates that were then used to assign the 'cryptic Alu-exons' (not annotated in UCSC as described above) as well as the 'alternative' or 'constitutive Alu-exon' based on overlapping coordinates. Genes containing more than one type of Alu-exon were assigned as 'constitutive' if they contained a constitutive Alu-exon or alternative if not. To calculate the median expression across tissues, we only considered genes with a median expression level of RPKM > 1, including 9,634 genes without an Alu-exon, as well as 78, 843 and 771 genes with a constitutive, alternative and cryptic Alu-exon, respectively. Average expression values for each tissue were calculated based only on genes expressed in the respective tissue (RPKM > 1).

## Statistics and software

Customised scripts written to characterise Alu-exons (such as PTC prediction, inclusion rate, U-tract lengths, liftover of Alu-exons to other species) are deposited at https://github.com/jernejule/Aluexonisation.

All statistical analyses were performed in the R software environment (version 2.15.3/3.1.3) or in GraphPad PRISM6. Whenever referred to in the text, *replicates* stands for biological replicates, defined as samples collected independently of one another in separated experiments. All experiments were done with biological replicates as indicated in Materials and methods and Figure legends, with the exception of the initial RNAseq experiment from cytoplasmic and nuclear RNA from cells depleted of hnRNPC combined with a time-course of RAF1 kinase induction, where a block design was used (see above).

Median gene expression values across human tissues were approximately normal distributed after $log_{10}$ transformation, as tested by the distribution of residuals (using the *car* package). Hence, significance testing for differential expression of Alu-exon genes across different tissues was performed

using $\log_{10}$-transformed data, with a one-way ANOVA design and Tukey's HSD to correct for multiple comparisons implemented in the *multcomp* package. All experiments using semi-quantitative RT-PCR were tested for significant changes in GraphPad PRISM, using a one-way ANOVA design and Tukey's HSD to correct for multiple comparisons. Data from cytoplasmic and nuclear RNA fractions were tested separately unless indicated otherwise. Differences in $\log_2$ fold changes of Alu-exon gene or exon abundances across groups were tested by two-sided Wilcoxon Rank Sum test or Kruskal-Wallis Rank Sum test using the *stats* or *pgirmess* packages. The association between splice site strength and Alu element substitution rate was tested by linear regression analysis using the *stats* package. For visualisation in *Figure 4A*, medians are presented with 95% confidence intervals estimated by bootstrapping with 2000 iterations and calculated with the Non-Studentised pivotal method. All bootstrappings were done with the *boot* package in R, using *boot()* and *boot.ci (type="basic")*.

## Supplemental dataset

List of exons with sufficient coverage for DEXSeq analysis in our RNAseq data. The supplementary dataset is the output of the DEXSeq analysis for RNAseq data set E-MTAB-6008. The table includes the position of each exon (hg19), Ensembl transcript and exon ID, base mean across conditions, and statistical testing scores by DEXSeq. Exon_class indicates if the exon is an Alu-exon, a cryptic exon, or not. UCSCoverlap and type indicate if the exon is annotated by UCSC, and if it is a constitutive or alternative exon. If the exon is not annotated in UCSC, it's called a cryptic exon. Stand_alone exons are not predicted by Cufflinks to overlap with another exon. The table also includes the number of junction-spanning reads confirming either splice site. In addition, we predicted premature stop codons in protein-coding genes; for PTC+ exons, it includes the distance between the PTC and downstream exon-exon junction, the following (second) exon-exon junction, and the number of downstream exons. The last two columns show the ID after merging all discovered Alu-exons with UCSC-annotated Alu-exons which can be cross-referenced with the source data table of *Figure 5*, and the grouping of evolutionary groups shown in *Figure 5*. The table is available from the Dryad data repository (accession doi:10.5061/dryad.7h81d).

## Acknowledgements

The authors thank all members of the Ule laboratory for assistance and discussion. We thank the Genomics Facility Team of the CRUK Cambridge Cancer Institute and the Genomics Unit at the Institute of Molecular Biology, Mainz for processing samples for high-throughput sequencing, especially Chung-Ting Han. We are grateful to Sarah K Jurmeister and Volker Böhm for input on this manuscript and critical comments. The HR1 cell line was a gift from Simon Cook (Babraham Institute, Cambridge). This work was supported by the European Research Council (grants 206726-CLIP and 617837-Translate to JU), and a Boehringer Ingelheim Fond PhD fellowship (to JA). KZ was supported by the LOEWE program Ubiquitin Networks (Ub-Net) of the State of Hesse (Germany).

## Additional information

### Funding

| Funder | Author |
| --- | --- |
| European Research Council | Jernej Ule |
| Boehringer Ingelheim Fonds | Jan Attig |
| LOEWE program Ubiquitin Networks of the State of Hesse | Kathi Zarnack |

The funders had no role in study design, data collection and interpretation, or the decision to submit the work for publication.

### Author contributions

JA, Conception and design, Acquisition of data, Analysis and interpretation of data, Drafting or revising the article; IRdlM, NH, WE, conducted data analyses and statistical testing, Analysis and

interpretation of data; ZW, generated polysome profiles, Acquisition of data; KZ, Analysis and interpretation of data, Drafting or revising the article; JK, Conception and design, Analysis and interpretation of data; JU, Conception and design, Analysis and interpretation of data, Drafting or revising the article

### Author ORCIDs

Jan Attig, http://orcid.org/0000-0002-2159-2880
Igor Ruiz de los Mozos, http://orcid.org/0000-0003-4097-6422
Jernej Ule, http://orcid.org/0000-0002-2452-4277

## Additional files

### Supplementary files

• Supplementary file 1. List of RT-PCR and RT-qPCR primers used in this study.

### Major datasets

The following datasets were generated:

| Author(s) | Year | Dataset title | Dataset URL | Database, license, and accessibility information |
|---|---|---|---|---|
| Attig J, Haberman N, Ruiz de los Mozos I, Wang Z, Zarnack K, König J, Ule J | 2016 | Transcriptome profiling of cells depleted of hnRNPC, UPF1, or co-depleted of hnRNPC and UPF1. | http://www.ebi.ac.uk/arrayexpress/experiments/E-MTAB-4008 | Publicly available at the EBI ArrayExpress (accession no: E-MTAB-4008) |
| Attig J, Haberman N, Ruiz de los Mozos I, Wang Z, Zarnack K, König J, Ule J | 2016 | Transcription profiling of cytoplasmic and nuclear RNA of hnRNPC-depleted cells and ER-RAF1 activated HR1 cells | http://www.ebi.ac.uk/arrayexpress/experiments/E-MTAB-4009 | Publicly available at the EBI ArrayExpress (accession no: E-MTAB-4009) |
| Attig J, Haberman N, Ruiz de los Mozos I, Wang Z, Zarnack K, König J, Ule J | 2016 | Data from: Splicing repression and NMD control the emergence of Alu-exons | http://dx.doi.org/10.5061/dryad.7h81d | Available at Dryad Digital Repository under a CC0 Public Domain Dedication |

The following previously published dataset was used:

| Author(s) | Year | Dataset title | Dataset URL | Database, license, and accessibility information |
|---|---|---|---|---|
| Schroth P | 2014 | RNA-Seq of human individual tissues and mixture of 16 tissues (Illumina Body Map) | http://www.ebi.ac.uk/arrayexpress/experiments/E-MTAB-513/ | Publicly available at the EBI ArrayExpress (accession no: E-MTAB-513) |

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
