## [Decision Letter]

Thank you for submitting your article "Splicing repression and NMD control the emergence of Alu-exons" for consideration by *eLife*. Your article has been favorably evaluated by James Manley (Senior Editor) and three reviewers, one of whom, Ben Blencowe (Reviewer #3), is a member of our Board of Reviewing Editors. The following individuals involved in review of your submission have agreed to reveal their identity: Manuel Irimia (Reviewer #1); Kristen W Lynch (Reviewer #2).

The reviewers have discussed the reviews with one another and the Reviewing Editor has drafted this decision to help you prepare a revised submission.

Summary:

In their submitted manuscript Ule and colleagues investigate mechanisms underlying the process of exonization that involve Alu repeat-derived sequences. Extending their previous work published in Cell demonstrating that hnRNPC has a widespread role in silencing Alu exons through binding to adjacent, cryptic 3´splice site polyU-tracts, the authors now show that nonsense mediated mRNA decay (NMD) can additionally act to suppress Alu exonization in a subset of transcripts, some of which are also suppressed by hnRNPC. However, a substantial proportion of Alu exon-containing transcripts that are predicted to be substrates for NMD do not appear to be targeted by this process, and many of these Alu exons are found adjacent to retained introns. The authors further show that ancient Alu exons are typified by having polyU tracts of reduced length but may often be associated with NMD activity, as supported by the observation that these transcripts have relatively reduced expression levels across human tissues.

Overall, this is an interesting and timely study that sheds light on mechanisms governing the suppression of potentially deleterious cryptic exons, the evolution of Alu-derived expressed exons, and a mechanism that may function to modulate mRNA expression across human tissues. Overall the authors' analyses are carefully performed and the conclusions mostly seem appropriate. However, it is requested that the authors address the following main points in a revised manuscript.

Essential revisions:

1) In the first part of the manuscript the authors deplete hnRNPC and assess, using RNA-Seq analysis of cytoplasmic and nuclear fractions, steady-state changes of transcripts that do – or do not – contain Alu exons. Differentially expressed transcripts are enriched for Alu-exons and there is an overall trend for down-regulation of cytoplasmic mRNA expression of genes that harbour these exons, which is validated for a subset of genes using RT-(q)PCR assays. However, hnRNPC depletion leads to differential expression of a large number of genes that lack Alu exons, indicating a more widespread role for hnRNPC in regulating steady-state mRNA levels independent of the presence of Alu-exons. To strengthen the conclusion that reduction in gene expression is specifically caused by increased Alu exon inclusion upon hnRNPC knockdown, rather than through alternative mechanisms, it would be useful to separately assess differential expression levels for those genes that contain cryptic, alternative, or constitutive Alu exons, where genes in the latter group should not be affected. Similarly, the authors should test whether there is a significant negative correlation between increased Alu exon inclusion and expression. This could be assessed in the UPF1-knockdown data where changes in splicing can be more accurately monitored.

2) Also related to point 1, an alternative explanation to the results of Figure 1 it that lowly expressed genes better tolerate Alu-exonization. In this regard, it would be informative to determine whether the expression of Alu-exon-free orthologs of these genes in mouse and other primates also have lower expression. If they do not, this would further support the authors' proposal.

3) The authors examine the role of NMD and show that two thirds of transcripts with hnRNPC-sensitive Alu exons are refractory to NMD (as assessed by UPF1 knockdown), even though the majority of these cases are computationally predicted to be NMD targets. The authors next show evidence that a significant fraction of these transcripts contain retained introns adjacent to the Alu-exons. They conclude that Alu-exon transcripts containing intron retention events can be readily exported and translated in the cytoplasm. However, their interpretation is based on RT-(q)PCR analysis of only a few transcripts, whereas it is unclear whether it applies more generally.

In particular, it is apparent that the majority of Alu exon-containing transcripts that are sensitive to hnRNPC knockdown show an increased nuclear:cytoplasmic ratio upon knockdown, while displaying reduced steady-state levels (lower left quadrant, above diagonal, in Figure 1—figure supplement 4 panel C). It is therefore recommended that the authors globally assess changes in the nuclear:cytoplasmic ratios of transcripts that include Alu-exons, with or without one or both adjacent retained introns, upon hnRNPC knockdown, using reads that span exon-intron junctions. While the authors' data may not be consistent with a "block" to nuclear RNA export that results in nuclear accumulation of Alu-exon transcripts as they claim, it is possible that intron retention in these transcripts leads to nuclear sequestration and subsequent nuclear turnover, and that such effects could in turn lead to reduced cytoplasmic levels of Alu-exon transcripts.

In relation to the above point, it is also unclear from the authors' data that retained intron-containing transcripts are associated with polysomes, beyond the one example that is analyzed. The authors should also not refer to polysome association as translation. Such a conclusion would require evidence at the peptide level to confirm that the corresponding protein is being produced.

4) The authors provide convincing data that Alu (and other) exons with flanking U-tracts are responsive to hnRNP C-depletion, but this doesn't rule out possible roles of other proteins. To what extent can the authors conclude that the U-tracts are functioning through hnRNP C versus altered recruitment of U2AF65 (further strengthening the splice sites), and/or allowing buffering by other U-tract binding hnRNP-like proteins (e.g. TIA). Some of the concern regarding the specificity of hnRNP C function is addressed by the author's earlier work, but it would be helpful to clarify this point in the text.

---

## [Author Response]

Overall, this is an interesting and timely study that sheds light on mechanisms governing the suppression of potentially deleterious cryptic exons, the evolution of Alu-derived expressed exons, and a mechanism that may function to modulate mRNA expression across human tissues. Overall the authors' analyses are carefully performed and the conclusions mostly seem appropriate. However, it is requested that the authors address the following main points in a revised manuscript.

Essential revisions:

1) In the first part of the manuscript the authors deplete hnRNPC and assess, using RNA-Seq analysis of cytoplasmic and nuclear fractions, steady-state changes of transcripts that do – or do not – contain Alu exons. Differentially expressed transcripts are enriched for Alu-exons and there is an overall trend for down-regulation of cytoplasmic mRNA expression of genes that harbour these exons, which is validated for a subset of genes using RT-(q)PCR assays. However, hnRNPC depletion leads to differential expression of a large number of genes that lack Alu exons, indicating a more widespread role for hnRNPC in regulating steady-state mRNA levels independent of the presence of Alu-exons. To strengthen the conclusion that reduction in gene expression is specifically caused by increased Alu exon inclusion upon hnRNPC knockdown, rather than through alternative mechanisms, it would be useful to separately assess differential expression levels for those genes that contain cryptic, alternative, or constitutive Alu exons, where genes in the latter group should not be affected. Similarly, the authors should test whether there is a significant negative correlation between increased Alu exon inclusion and expression. This could be assessed in the UPF1-knockdown data where changes in splicing can be more accurately monitored.

The reviewers suggested valuable additional analysis that we included in our revised manuscript.

Following the reviewers’ suggestion we distinguished Alu-exon genes according to the type of Alu-exon (cryptic/alternative/constitutive; with minimal coverage of 5 reads to ensure sufficient inclusion) and separately assessed their differential expression levels. Indeed, constitutive Alu-exons do not lead to loss of gene expression after hnRNPC depletion, while cryptic Alu-exons do. We included the results as new Figure 1.

We also tested whether Alu-exons with high inclusion correlate with stronger reduction in gene expression. The reviewers correctly pointed out that inclusion of Alu-exons will be highest in UPF1-depleted cells, and we hence ranked Alu-exons according to their fold change in abundance in our hnRNPC/UPF1 co-depletion conditions, which should measure the highest inclusion each exon can achieve. Notably, increased Alu-exon usage correlates with stronger reduction of the corresponding gene in hnRNPC-depleted cells (Spearman rank correlation, p-value < 2.2e^-16^). We included the results as new Figure 2.

We believe that both figure panels solidify one of our central arguments, namely that inclusion of Alu-exons renders the transcripts susceptible to NMD, and that this explains to considerable extent the changes in gene expression in hnRNPC-depleted cells.

2) Also related to point 1, an alternative explanation to the results of Figure 1 it that lowly expressed genes better tolerate Alu-exonization. In this regard, it would be informative to determine whether the expression of Alu-exon-free orthologs of these genes in mouse and other primates also have lower expression. If they do not, this would further support the authors' proposal.

We thank the reviewers for pointing out this alternative explanation for our initial observation of reduced expression of Alu-exon genes. We agree that it is possible that lowly expressed genes could show less negative selection against Alu-exonisation events, which would make them appear more tolerant. However, this also implies that Alu-exonisation is detrimental to gene expression levels. If we assume highly expressed genes are not tolerant to Alu-exonisation, but lowly expressed genes are, this would indicate a negative selection acting on highly expressed genes. This is presumably because Alu-exonisation events hamper expression or function of those genes, and high/robust gene expression is more frequently associated with essential cellular or tissue-specific function (1, 2). We modified the respective section in the text to acknowledge both scenarios, which are not mutually exclusive:

“The reduced expression of Alu-exon genes might result from a combination of two scenarios. Genes with low expression might be less essential to the organism, or evolve more quickly, which would decrease negative selection against inclusion of Alu-exons. Alternatively, inclusion of Alu-exons could be the primary cause of the low gene expression as a result of quality control pathways that degrade Alu-exon transcripts.”

As suggested, we did compare expression of human Alu-exon genes and their orthologues across mammalian species in publicly available data sets. One such data set from Brawand et al. (Nature 2011) sequenced RNA from six organs across ten species (including human, mouse, opossum, platypus, chicken and 4 non-human primates). In human, the data reproduced the trend we report in Figure 1 (shown in Figure 7).

Regarding the comparison to other species, we first tested the underlying assumption that the genes expressed at the lowest level in human are generally also lowly expressed in other species. We observed an increase in gene expression and expression variance roughly correlating with evolutionary distance (shown in Figure 7), and concluded that only comparisons up to mouse are valid. Analysis of the Alu-exon genes showed that orthologues of the human Alu-exon genes show no expression bias in mouse but do so in primates, which share some of the Alu-exons with human, consistent with the conclusion that Alu-exons could reduce gene expression. Yet we rely on 1-to-1 orthologue mapping which reduces the number of data points (genes) in each group substantially; for instance only 30 out of 120 human genes with constitutive Alu-exons are found across species.

Due to the small numbers and confounding factors within the cross-species gene expression patterns, we decided not to include these analyses in the manuscript.

Author response image 1.(**A**) We selected the ~15% most lowly expressed genes across six human tissues (RPKM of 0.5-2, n=2,199), and compared their expression levels (RPKM) in other species. Data from Brawand et al. (Nature 2011). (**B**) We compared expression levels (RPKM) of Alu-exon genes and their mammalian orthologues in primates and mouse. Species abbreviated as in Figure 1.**DOI:**
http://dx.doi.org/10.7554/eLife.19545.023

References cited above:

1) Hart T, Brown KR, Sircoulomb F, Rottapel R, Moffat J. Measuring error rates in genomic perturbation screens: gold standards for human functional genomics. Mol Syst Biol. 2014 Jul 01;10:733. PubMed PMID: 24987113. Pubmed Central PMCID: 4299491.

2) Wang T, Birsoy K, Hughes NW, Krupczak KM, Post Y, Wei JJ, et al. Identification and characterization of essential genes in the human genome. Science. 2015 Nov 27;350(6264):1096-101. PubMed PMID: 26472758. Pubmed Central PMCID: 4662922.

*3) The authors examine the role of NMD and show that two thirds of transcripts with hnRNPC-sensitive Alu exons are refractory to NMD (as assessed by UPF1 knockdown), even though the majority of these cases are computationally predicted to be NMD targets. The authors next show evidence that a significant fraction of these transcripts contain retained introns adjacent to the Alu-exons. They conclude that Alu-exon transcripts containing intron retention events can be readily exported and translated in the cytoplasm. However, their interpretation is based on RT-(q)PCR analysis of only a few transcripts, whereas it is unclear whether it applies more generally.*

*In particular, it is apparent that the majority of Alu exon-containing transcripts that are sensitive to hnRNPC knockdown show an increased nuclear:cytoplasmic ratio upon knockdown, while displaying reduced steady-state levels (lower left quadrant, above diagonal, in Figure 1—figure supplement 4 panel C). It is therefore recommended that the authors globally assess changes in the nuclear:cytoplasmic ratios of transcripts that include Alu-exons, with or without one or both adjacent retained introns, upon hnRNPC knockdown, using reads that span exon-intron junctions. While the authors' data may not be consistent with a "block" to nuclear RNA export that results in nuclear accumulation of Alu-exon transcripts as they claim, it is possible that intron retention in these transcripts leads to nuclear sequestration and subsequent nuclear turnover, and that such effects could in turn lead to reduced cytoplasmic levels of Alu-exon transcripts.*

In relation to the above point, it is also unclear from the authors' data that retained intron-containing transcripts are associated with polysomes, beyond the one example that is analyzed. The authors should also not refer to polysome association as translation. Such a conclusion would require evidence at the peptide level to confirm that the corresponding protein is being produced.

We acknowledge the reviewers’ criticism that the polysome association is limited to the exemplary transcripts analysed by RT-(q)PCR and should generally not be taken as direct evidence of translation, and have amended the text to address this. To this end, we included in the Discussion the following sentences:

“We observed that NMD-resistant Alu-exon transcripts of *TNPO3* and *ZFX* as well as the intron-retaining Alu transcript of *NUP133* are associated with polysomes in a similar ratio as the conventionally spliced transcripts. […] Moreover, it will be exciting to see whether the polysome-associated intron transcripts are indeed being translated, for instance by mining deep proteomics data collected from hnRNPC-depleted cells.”

With respect to the notion that intron associated transcripts might be retained in the nucleus (coupled to enhanced decay), we first attempted to use exon-intron spanning reads as suggested by the reviewer. However, since the read numbers on many Alu-exon:intron junctions were low, signal-to-noise ratio was too poor to draw firm conclusions from the analysis. Instead, we used all reads within introns of a gene and all exon-exon junction reads und assessed the ratio of spliced to unspliced transcripts. We separated non-Alu-exons genes versus Alu-exon genes with or without significant amounts of Alu-associated intron transcripts. Within the nuclear RNAseq data set, we find no appreciable accumulation of unspliced RNA (vs spliced transcripts) in either group upon hnRNPC depletion. Together with our more extensive RT-PCR evidence that the intron-retained Alu transcripts of NUP133, GMPS and MCM3 are by large not retained in the nucleus, we believe that this argues against a general bias of alternatively spliced transcripts to be sequestered in the nucleus.

4) The authors provide convincing data that Alu (and other) exons with flanking U-tracts are responsive to hnRNP C-depletion, but this doesn't rule out possible roles of other proteins. To what extent can the authors conclude that the U-tracts are functioning through hnRNP C versus altered recruitment of U2AF65 (further strengthening the splice sites), and/or allowing buffering by other U-tract binding hnRNP-like proteins (e.g. TIA). Some of the concern regarding the specificity of hnRNP C function is addressed by the author's earlier work, but it would be helpful to clarify this point in the text.

We appreciate the reviewers’ point that the specific connection between U-tracts and hnRNPC vs. other splicing regulators was not sufficiently covered in the text, and have addressed this in the revised manuscript.

We included a section in the Introduction to explain that no other splicing regulator tested in the past has shown increased inclusion of a multitude of Alu-exons:

“In principle, it is plausible that other proteins with known preference for U-rich motifs (e.g. HuR, TIA, TDP43) also bind the Alu U-tract and repress Alu exonisation; and indeed HuR and TDP43 show enriched binding at antisense Alu elements (Kelley et al., 2014). Yet, depletion experiments for TDP43, TIA, HuR, PTB and hnRNPA1 did not show increased Alu-exon inclusion in their absence (Zarnack et al., 2013; Kelley et al., 2014). Thus, the U-tract:hnRNPC interaction is crucial to prevent the splicing machinery from accessing cryptic splice sites at Alu-exons.”

We also speculate in the Discussion about a ‘hand-over’ scenario for individual Alu-exons, in which universal repression by hnRNPC across tissues is replaced by a tissue-specific splice regulator – and acknowledge that this is not addressed in the manuscript and remains hypothetical.

“However, it might be far more likely that the regulatory elements diversify. […] Since it is likely that each tissue-specific Alu-exon is regulated by a unique combination of RBPs, it will require further studies to fully unravel these more complex regulatory networks.